# Error Compensated Distributed SGD
# can be Accelerated

**Xun Qian**[*]
xun.qian@kaust.edu.sa

**Peter Richtárik**[†]
peter.richtarik@kaust.edu.sa

**Tong Zhang**[‡]
tongzhang@ust.hk

## Abstract

Gradient compression is a recent and increasingly popular technique for reducing the communication cost in distributed training of large-scale machine learning models. In this work we focus on developing efficient distributed methods that can work for any compressor satisfying a certain contraction property, which includes both unbiased (after appropriate scaling) and biased compressors such as RandK and TopK. Applied naively, gradient compression introduces errors that either slow down convergence or lead to divergence. A popular technique designed to tackle this issue is error compensation/error feedback. Due to the difficulties associated with analyzing biased compressors, it is not known whether gradient compression with error compensation can be combined with acceleration. In this work, we show for the first time that error compensated gradient compression methods can be accelerated. In particular, we propose and study the error compensated loopless Katyusha method, and establish an accelerated linear convergence rate under standard assumptions. We show through numerical experiments that the proposed method converges with substantially fewer communication rounds than previous error compensated algorithms.

## 1 Introduction

When training very large scale supervised machine learning problems, such as those arising in the context of federated learning [Konečný et al., 2016b, McMahan et al., 2017, Konečný et al., 2016a] (see also recent surveys [Li et al., 2019, Kairouz, 2019]), distributed algorithms are indispensable. In such settings, communication is generally much slower than (local) computation, which makes it the key bottleneck in the design of efficient distributed systems. There are several ways to tackle this issue, including reliance on large mini-batches [Goyal et al., 2017, You et al., 2017], asynchronous learning [Agarwal and Duchi, 2011, Lian et al., 2015, Recht et al., 2011], local updates [Ma et al., 2017, Stich, 2020, Khaled et al., 2020, Hanzely and Richtárik, 2020, Woodworth et al., 2020] and communication compression (e.g., quantization and sparsification) [Alistarh et al., 2017, Bernstein et al., 2018, Mishchenko et al., 2019, Seide et al., 2014, Wen et al., 2017].

**Communication compression.** In this work, we focus on the last of these techniques: *communication compression*. The key idea here is to apply a lossy compression transformation/operator to the messages before they are communicated so as to save on communication time. While compression reduces the communicated bits in each communication round, it introduces errors, and this generally leads to an increase in the number of communication rounds needed to find a solution of any predefined accuracy. Still, compression has been found useful in practice, as the trade-off often seems to prefer compression to no compression.

---

[*]King Abdullah University of Science and Technology, Thuwal, Saudi Arabia

[†]King Abdullah University of Science and Technology, Thuwal, Saudi Arabia; Moscow Institute of Physics and Technology, Dolgoprudny, Russia

[‡]Hong Kong University of Science and Technology, Hong Kong

35th Conference on Neural Information Processing Systems (NeurIPS 2021).

**Contractive and unbiased compressors.** There are two large families of such compression operators: *contraction* and *unbiased* compressors [Beznosikov et al., 2020]. A (possibly) randomized map $Q : \mathbb{R}^d \to \mathbb{R}^d$ is called a *contraction compressor* if there exists a constant $0 < \delta \leq 1$ such that

$$\mathbb{E}\left[\|x - Q(x)\|^2\right] \leq (1 - \delta)\|x\|^2, \qquad \forall x \in \mathbb{R}^d. \tag{1}$$

Further, we say that a randomized map $\tilde{Q} : \mathbb{R}^d \to \mathbb{R}^d$ is an *unbiased compressor* if there exists a constant $\omega \geq 0$ such that

$$\mathbb{E}\left[\tilde{Q}(x)\right] = x \quad \text{and} \quad \mathbb{E}\left[\|\tilde{Q}(x)\|^2\right] \leq (\omega + 1)\|x\|^2, \qquad \forall x \in \mathbb{R}^d. \tag{2}$$

It is well known that (see, e.g., [Beznosikov et al., 2020]) after appropriate scaling, any unbiased compressor satisfying (2) becomes a contraction compressor. Indeed, it is easy to verify that for any $\tilde{Q}$ satisfying (2), $\frac{1}{\omega+1}\tilde{Q}$ is a contraction compressor satisfying (1) with $\delta = 1/(\omega+1)$. Thus we can construct corresponding contraction compressor from any unbiased compressor. However, there are empirically very powerful contractive compressors, such as TopK [Alistarh et al., 2018] (described below), which do not arise this way, and because of this, the class of contractive compressors is important in practice. For illustration purposes, we now define two canonical examples of contraction compressors. Let $1 \leq K \leq d$. The TopK compressor is defined as

$$(\text{TopK}(x))_{\pi(i)} = \left\{ \begin{array}{ll} (x)_{\pi(i)} & \text{if } i \leq K, \\ 0 & \text{otherwise,} \end{array} \right.$$

where $\pi$ is a permutation of $\{1, 2, ..., d\}$ such that $(|x|)_{\pi(i)} \geq (|x|)_{\pi(i+1)}$ for $i = 1, ..., d - 1$. The RandK compressor is defined as

$$(\text{RandK}(x))_i = \left\{ \begin{array}{ll} (x)_i & \text{if } i \in S, \\ 0 & \text{otherwise,} \end{array} \right.$$

where $S$ is chosen uniformly from the set of all $K$ element subsets of $\{1, 2, ..., d\}$. For TopK and RandK compressors, we have $\delta \geq K/d$ [Stich et al., 2018]. Some frequently used unbiased compressors include random dithering [Alistarh et al., 2017], random sparsification [Stich et al., 2018], and natural compression [Horváth et al., 2019b]. For more examples of contraction and unbiased compressors, we refer the reader to [Beznosikov et al., 2020].

**Related Work.** If we assume that the accumulated error is bounded, and in the case of unbiased compressors, the convergence rate of error compensated SGD was shown to be the same as that of vanilla SGD [Tang et al., 2018]. However, if we only assume bounded second moment of the stochastic gradients, in order to guarantee the boundedness of the accumulated quantization error, some decaying factors need to be involved in general, and error compensated SGD is proved to have some advantage over QSGD [Alistarh et al., 2017] in some perspective for convex quadratic problems [Wu et al., 2018]. On the other hand, for contraction compressors, error compensated SGD actually has the same convergence rate as vanilla SGD [Stich et al., 2018, Stich and Karimireddy, 2019, Tang et al., 2019]. Since SGD only has a sublinear convergence rate, these error compensated methods could not get linear convergence rate. If $f$ is non-smooth and $\psi = 0$, error compensated SGD was studied in [Karimireddy et al., 2019] in the single node case, and the convergence rate is of order $O\left(1/\sqrt{\delta k}\right)$.

For variance-reduced methods, QSVRG [Alistarh et al., 2017] handles the smooth case ($\psi \equiv 0$), and VR-DIANA [Horváth et al., 2019a], ADIANA [Li et al., 2020] handle the composite case (general $\psi$). However, the compressors of these algorithms need to be unbiased. Error compensation in VR-DIANA and ADIANA does not need to be used since their method successfully employs variance reduction (of the variance introduced by the compressor) instead. Recently, an error compensated method called EC-LSVRG-DIANA which can achieve nonaccelerated linear convergence for the strongly convex and smooth case was proposed in [Gorbunov et al., 2020], but besides the contraction compressor, the unbiased compressor is also needed in the algorithm.

**Error compensation.** While contractive compressors can be more powerful in practice, they are not unbiased, and this leads to severe difficulties in the algorithm design space, and poses challenges for theoreticians trying to understand the convergence behavior of distributed algorithms using contractive compressors. Indeed, while there is no issue with the use of contractive compressors in the non-distributed setting (which is of course not interesting from a practical point of view) in

combination with standard gradient-type methods, serious convergence issues arise in the distributed setting. Indeed, classical methods such gradient descent can diverge (even exponentially fast) when the TopK compressor is used to compress uplink gradient information [Beznosikov et al., 2020]. Fortunately, this issue can be resolved using the *error compensation (EC)*[4] mechanism, which can be traced back to at least Seide et al. [2014]. However, this mechanism remained a heuristic until recently. To highlight the difficulty of analyzing EC, we remark that the first theoretical breakthroughs applied to the single node regime only [Stich et al., 2018, Stich and Karimireddy, 2019]. More recently, theoretical progress was made in the crucially important distributed setting as well [Beznosikov et al., 2020, Gorbunov et al., 2020]. However, we are still very far in our understanding of optimization algorithms using contractive compressors.

> *One of the key open problems in this area—the problem we address in this paper in the affirmative— is whether it is possible to design provably accelerated gradient-type methods that work with contractive compressors.*

In this paper, we give a confirmed answer by studying error compensation in conjunction with the acceleration mechanism employed in loopless Katyusha (L-Katyusha) [Kovalev et al., 2020, Qian et al., 2019], which is a loopless variant of Katyusha [Allen-Zhu, 2017]. The acceleration in Katyusha or L-Katyusha is an extension of Nesterov's acceleration [Nesterov, 1983, 2004] in the stochastic regime. Moreover, Katyusha and L-Katyusha can achieve the optimal iteration complexity among randomized methods for solving finite-sum strongly convex optimization problems [Lan and Zhou, 2018].

## 2 Problem Description and Summary of Contributions

We formally describe the distributed optimization problem we solve in Section 2.1, and then summarize our key contributions in Section 2.2.

### 2.1 Problem Description

We consider the composite finite-sum optimization problem

$$\min_{x \in \mathbb{R}^d} P(x) := \left\{ \frac{1}{n} \sum_{\tau=1}^{n} f^{(\tau)}(x) + \psi(x) \right\}, \tag{3}$$

where $f(x) := \frac{1}{n} \sum_{\tau} f^{(\tau)}(x)$ is an average of $n$ smooth convex functions, distributed over $n$ compute nodes, and $\psi : \mathbb{R}^d \to \mathbb{R} \cup \{+\infty\}$ is a proper closed convex function representing a possibly nonsmooth regularizer. On each node $\tau$, $f^{(\tau)}(x)$ is an average of $m$ smooth convex functions, i.e., $f^{(\tau)}(x) = \frac{1}{m} \sum_{i=1}^{m} f_i^{(\tau)}(x)$, representing the average loss over the training data stored on this node.

We denote the smoothness constants of functions $f$, $f^{(\tau)}$ and $f_i^{(\tau)}$ using symbols $L_f$, $\bar{L}$ and $L$, respectively.[5] These constants are in general related as follows:

$$L_f \leq \bar{L} \leq n L_f, \qquad \bar{L} \leq L \leq m\bar{L}. \tag{4}$$

While we specifically focus on the case when $m = 1$, our results are also new in the $m = 1$ case, and hence this regime is relevant as well. We assume throughout that problem (3) has at least one optimal solution $x^*$.

We will optionally use the following additional assumption for the contraction compressor. The assumption is not necessary, but when used, it will lead to better complexity.

**Assumption 2.1.** $\mathbb{E}[Q(x)] = \delta x$ and $\mathbb{E}[Q_1(x)] = \delta_1 x$ for all $x \in \mathbb{R}^d$, where $\delta$ and $\delta_1$ are compression parameters of $Q$ and $Q_1$, respectively.

It is easy to verify that RandK compressor satisfies Assumption 2.1 with $\delta = K/d$, and $\frac{1}{\omega+1}\tilde{Q}$, where $\tilde{Q}$ is any unbiased compressor, also satisfies Assumption 2.1 with $\delta = 1/(\omega+1)$.

---

[4]Error compensation is also known under the name error feedback.

[5]Function $f : \mathbb{R}^d \to \mathbb{R}$ is smooth if it is differentiable, and has $L$-Lipschitz gradient: $\|\nabla f(x) - \nabla f(y)\| \leq L\|x - y\|$ for any $x, y \in \mathbb{R}^d$. $L$ is called the *smoothness constant* of $f$.

## 2.2 Contributions

We now summarize the main contributions of our work.

**Acceleration for error compensation.** We develop a new communication efficient algorithm (Algorithm 1) for solving the distributed optimization problem (3) which we call *Error Compensated Loopless Katyusha* (ECLK). ECLK is the *first accelerated* error compensated method (this is true even in the non-distributed setting), which we believe is a significant contribution to the field.

**Iteration complexity.** We obtain the *first accelerated linear convergence rate* for error compensated methods using contraction operators. Let $p \leq \mathcal{O}(\delta_1)$, where $p \in (0, 1]$ is a parameter of the method described later and $\delta_1$ is the compression parameter of $Q_1$ in Algorithm 1 (see Assumption 2.1). Then the iteration complexity of ECLK is

$$\mathcal{O}\left(\left(\frac{1}{\delta} + \frac{1}{p} + \sqrt{\frac{L_f}{\mu}} + \sqrt{\frac{L}{\mu p n}} + \frac{1}{\delta}\sqrt{\frac{(1-\delta)\bar{L}}{\mu p}} + \sqrt{\frac{(1-\delta)L}{\mu p \delta}}\right) \log \frac{1}{\epsilon}\right). \qquad (5)$$

This is an improvement over the previous best known results for error compensated methods in [Beznosikov et al., 2020] and [Gorbunov et al., 2020], where *nonaccelerated* linear rates were obtained for the smooth case ($\psi \equiv 0$). Moreover, in oder to obtained a linear rate, Beznosikov et al. [2020] required the assumption that $\nabla f^{(\tau)}(x^*) = 0$ for all $\tau$, and also need full gradients to be computed by all nodes. The complexity of EC-LSVRG-DIANA in [Gorbunov et al., 2020] is

$$\mathcal{O}\left(\left(\omega + m + \frac{L}{\delta\mu}\right) \log \frac{1}{\epsilon}\right). \qquad (6)$$

When the last term in (6) is dominant and the last two terms in (5) are dominant, the complexity of ECLK is better than that of EC-LSVRG-DIANA if $p \geq \mathcal{O}\left(\frac{(1-\delta)\bar{L}\mu}{L^2} + \frac{(1-\delta)\delta\mu}{L}\right)$.

If we invoke additional assumptions (Assumption 2.1) on the contraction compressor, the iteration complexity of ECLK is improved to

$$\mathcal{O}\left(\left(\frac{1}{\delta} + \frac{1}{p} + \sqrt{\frac{L_f}{\mu}} + \sqrt{\frac{L}{\mu p n}} + \frac{1}{\delta}\sqrt{\frac{(1-\delta)L_f}{\mu p}} + \sqrt{\frac{(1-\delta)L}{\mu p \delta n}}\right) \log \frac{1}{\epsilon}\right).$$

This is indeed an improvement since $L_f \leq \bar{L}$ (see (4)), and because of the extra scaling factor of $n$ in the last term. If $\delta = \delta_1 = 1$, i.e., if no compression is used, we recover the iteration complexity of the accelerated method L-Katyusha [Qian et al., 2019].

# 3 Error compensated L-Katyusha

## 3.1 Description of the method

In this section we describe our method: error compensated L-Katyusha (see Algorithm 1). Roughly speaking, ECLK is a combination of L-Katyusha, error feedback, and a learning scheme in VR-DIANA [Horváth et al., 2019a].

The search direction in L-Katyusha in the distributed setting ($n \geq 1$) at iteration $k$ is

$$\frac{1}{n}\sum_{\tau=1}^{n}\left(\nabla f_{i_k^\tau}^{(\tau)}(x^k) - \nabla f_{i_k^\tau}^{(\tau)}(w^k) + \nabla f^{(\tau)}(w^k)\right),$$

where $i_k^\tau$ is sampled uniformly and independently from $[m] := \{1, 2, ..., m\}$ on the $\tau$-th node for $1 \leq \tau \leq n$, $x^k$ is the current iteration, and $w^k$ is the current reference point. Whenever $\psi$ is nonzero in problem (3), $\nabla f(x^*)$ is nonzero in general, and so is $\nabla f^{(\tau)}(x^*)$ (it could be nonzero even if $\nabla f(x^*) = 0$ in the non-regularized case). Thus, compressing the direction

$$\nabla f_{i_k^\tau}^{(\tau)}(x^k) - \nabla f_{i_k^\tau}^{(\tau)}(w^k) + \nabla f^{(\tau)}(w^k) \qquad (7)$$

directly on each node would cause nonzero noise even if $x^k$ and $w^k$ converged to the optimal solution $x^*$. On the other hand, since $f_i^{(\tau)}$ is $L$-smooth, $\nabla f_{i_k^\tau}^{(\tau)}(x^k) - \nabla f_{i_k^\tau}^{(\tau)}(w^k)$ could be small if $x^k$ and $w^k$ are close enough. For the last term $\nabla f^{(\tau)}(w^k)$, we introduce a vector $h_\tau^k$ to learn it iteratively in

a similar way in VR-DIANA. However, we use a contraction compressor rather than an unbiased one. More precisely, we take the update

$$h_\tau^{k+1} = h_\tau^k + Q_1(\nabla f^{(\tau)}(w^k) - h_\tau^k), \tag{8}$$

where $Q_1$ is a contraction compressor. $h_\tau^0$ can be initialized by any vector in $\mathbb{R}^d$. It is possible to interpret the learning procedure (8) as one step of the compressed gradient descent method applied to a certain quadratic optimization problem whose unique solution is the vector $\nabla f^{(\tau)}(w^k)$. Now we would expect $\nabla f^{(\tau)}(w^k) - h_\tau^k$ could converge to zero as $k \to +\infty$. We substract $h_\tau^k$ from the search direction in (7) to get

$$g_\tau^k = \nabla f_{i_k^\tau}^{(\tau)}(x^k) - \nabla f_{i_k^\tau}^{(\tau)}(w^k) + \nabla f^{(\tau)}(w^k) - h_\tau^k,$$

and will add $h_\tau^k$ back after aggregation.

Next we apply the compression and error feedback techniques, i.e., we compress the vector $\frac{\eta}{\mathcal{L}_1} g_\tau^k + e_\tau^k$ on each node to get $\tilde{g}_\tau^k = Q(\frac{\eta}{\mathcal{L}_1} g_\tau^k + e_\tau^k)$, where $Q$ is a contraction compressor and $\mathcal{L}_1$ is a positive parameter. The accumulated error $e_\tau^{k+1}$ is updated by the compression error at iteration $k$. On each node, a scalar $u_\tau^k$ is also maintained, and only $u_1^k$ will be updated. The summation of $u_\tau^k$ for $1 \le \tau \le n$ is $u^k$, and we use $u^k$ to control the update frequency of the reference point $w^k$.[6] Each node sends the two compressed vectors $\tilde{g}_\tau^k$, $Q_1(\nabla f^{(\tau)}(w^k) - h_\tau^k)$, and a scalar $u_\tau^{k+1}$ to the other nodes. After aggregating $\tilde{g}^k = \frac{1}{n}\sum_{\tau=1}^n \tilde{g}_\tau^k$, we add the average vector $h^k = \frac{1}{n}\sum_{\tau=1}^n h_\tau^k$ after multiplying the stepsize $\frac{\eta}{\mathcal{L}_1}$ to it to get the search direction $\tilde{g}^k + \frac{\eta}{\mathcal{L}_1} h^k$. We also use the following standard proximal operator for the update of $z^k$:

$$\text{prox}_{\eta\psi}(x) := \arg\min_y \left\{ \tfrac{1}{2}\|x - y\|^2 + \eta\psi(y) \right\}.$$

The reference point $w^k$ will be updated if $u^{k+1} = 1$. It is easy to see that $w^k$ will be updated with probability $p$ at each iteration. All nodes maintain the same copies of $x^k$, $w^k$, $y^k$, $z^k$, $\tilde{g}^k$, $h^k$, and $u^k$.

We need the following assumptions in this section.

**Assumption 3.1.** *The two compressors $Q$ and $Q_1$ in Algorithm 1 are contraction compressors with parameters $\delta$ and $\delta_1$, respectively.*

**Assumption 3.2.** *$f_i^{(\tau)}$ is $L$-smooth, $f^{(\tau)}$ is $\bar{L}$-smooth, $f$ is $L_f$-smooth and $\mu_f$-strongly convex, and $\psi$ is $\mu_\psi$-strongly convex, where $\mu_f \ge 0$, $\mu_\psi \ge 0$, and $\mu := \mu_f + \mu_\psi > 0$.*

### 3.2 Convergence analysis: preliminaries

First, we introduce some perturbed vectors which will be used in the convergence analysis. The perturbed vector is usually used in the analysis of error compensated methods [Karimireddy et al., 2019]. In Algorithm 1, let $e^k = \frac{1}{n}\sum_{\tau=1}^n e_\tau^k$, $g^k = \frac{1}{n}\sum_{\tau=1}^n g_\tau^k$, and $\tilde{x}^k = x^k - \frac{1}{1+\eta\sigma_1}e^k$, $\tilde{z}^k = z^k - \frac{1}{1+\eta\sigma_1}e^k$ for $k \ge 0$. Then $e^{k+1} = \frac{1}{n}\sum_{\tau=1}^n \left( e_\tau^k + \frac{\eta}{\mathcal{L}_1}g_\tau^k - \tilde{g}_\tau^k \right) = e^k + \frac{\eta}{\mathcal{L}_1}g^k - \tilde{g}^k$, and from the optimality condition for the proximal operator, we have

$$
\begin{aligned}
\tilde{z}^{k+1} &= z^{k+1} - \tfrac{1}{1+\eta\sigma_1}e^{k+1} \\
&= \tfrac{1}{1+\eta\sigma_1}\left( \eta\sigma_1 x^k + z^k - \tilde{g}^k - \tfrac{\eta}{\mathcal{L}_1}h^k \right) - \tfrac{\eta\partial\psi(z^{k+1})}{(1+\eta\sigma_1)\mathcal{L}_1} - \tfrac{e^{k+1}}{1+\eta\sigma_1} \\
&= \tfrac{1}{1+\eta\sigma_1}\left( \eta\sigma_1 x^k + z^k - e^k - \tfrac{\eta}{\mathcal{L}_1}g^k - \tfrac{\eta}{\mathcal{L}_1}h^k \right) - \tfrac{\eta\partial\psi(z^{k+1})}{(1+\eta\sigma_1)\mathcal{L}_1} \\
&= \tfrac{1}{1+\eta\sigma_1}\left( \eta\sigma_1 \tilde{x}^k + \tilde{z}^k - \tfrac{\eta}{\mathcal{L}_1}g^k - \tfrac{\eta}{\mathcal{L}_1}h^k \right) - \tfrac{\eta\partial\psi(z^{k+1})}{(1+\eta\sigma_1)\mathcal{L}_1}. \tag{9}
\end{aligned}
$$

The above relation plays a vital role in the convergence analysis, and allows us to follow the analysis of original L-Katyusha. In particular, we will use $\tilde{z}^k$ to construct Lyapunov functions. Next we define some notations which will be used to construct Lyapunov functions. Define $\tilde{\mathcal{Z}}^k := \frac{\mathcal{L}_1 + \eta\mu/2}{2\eta}\|\tilde{z}^k -$

---

[6]We can also use a single shared random variable $u^k$ instead.

**Algorithm 1** Error Compensated Loopless Katyusha (ECLK)

---

1: **Parameters:** stepsize parameters $\eta = \frac{1}{3\theta_1} > 0$, $\mathcal{L}_1 > 0$, $\sigma_1 = \frac{\mu_f}{2\mathcal{L}_1} \geq 0$, $\theta_1, \theta_2 \in (0,1)$; probability $p \in (0,1]$
2: **Initialization:** $x^0 = y^0 = z^0 = w^0 \in \mathbb{R}^d$; $u_\tau^0 = 0 \in \mathbb{R}$; $e_\tau^0 = 0 \in \mathbb{R}^d$; $h_\tau^0 \in \mathbb{R}^d$; $u^0 = \sum_{\tau=1}^n u_\tau^0$; $h^0 = \frac{1}{n}\sum_{\tau=1}^n h_\tau^0$
3: **for** $k = 0, 1, 2, \dots$ **do**
4:    **for** $\tau = 1, \dots, n$ **do**
5:       Sample $i_k^\tau$ uniformly and independently in $[m]$ on each node
6:       $g_\tau^k = \nabla f_{i_k^\tau}^{(\tau)}(x^k) - \nabla f_{i_k^\tau}^{(\tau)}(w^k) + \nabla f^{(\tau)}(w^k) - h_\tau^k$
7:       $\tilde{g}_\tau^k = Q(\frac{\eta}{\mathcal{L}_1}g_\tau^k + e_\tau^k)$,   $h_\tau^{k+1} = h_\tau^k + Q_1(\nabla f^{(\tau)}(w^k) - h_\tau^k)$
8:       $e_\tau^{k+1} = e_\tau^k + \frac{\eta}{\mathcal{L}_1}g_\tau^k - \tilde{g}_\tau^k$,   $u_\tau^{k+1} = 0$ for $\tau = 2, \dots, n$
9:       $u_1^{k+1} = \begin{cases} 1 & \text{with probability } p \\ 0 & \text{with probability } 1 - p \end{cases}$
10:      Send $\tilde{g}_\tau^k$, $Q_1(\nabla f^{(\tau)}(w^k) - h_\tau^k)$, and $u_\tau^{k+1}$ to the other nodes
11:      Receive $\tilde{g}_\tau^k$, $Q_1(\nabla f^{(\tau)}(w^k) - h_\tau^k)$, and $u_\tau^{k+1}$ from the other nodes
12:      $\tilde{g}^k = \frac{1}{n}\sum_{\tau=1}^n \tilde{g}_\tau^k$,   $u^{k+1} = \sum_{\tau=1}^n u_\tau^{k+1}$
13:      $z^{k+1} = \text{prox}_{\frac{\eta}{(1+\eta\sigma_1)\mathcal{L}_1}\psi}\left(\frac{1}{1+\eta\sigma_1}\left(\eta\sigma_1 x^k + z^k - \tilde{g}^k - \frac{\eta}{\mathcal{L}_1}h^k\right)\right)$
14:      $y^{k+1} = x^k + \theta_1(z^{k+1} - z^k)$
15:      $w^{k+1} = \begin{cases} y^k & \text{if } u^{k+1} = 1 \\ w^k & \text{otherwise} \end{cases}$
16:      $x^{k+1} = \theta_1 z^{k+1} + \theta_2 w^{k+1} + (1 - \theta_1 - \theta_2)y^{k+1}$
17:      $h^{k+1} = h^k + \frac{1}{n}\sum_{\tau=1}^n Q_1(\nabla f^{(\tau)}(w^k) - h_\tau^k)$
18:    **end for**
19: **end for**

---

$x^*\|^2$, $\mathcal{Y}^k := \frac{1}{\theta_1}(P(y^k) - P^*)$, and $\mathcal{W}^k := \frac{\theta_2}{pq\theta_1}(P(w^k) - P^*)$ for $k \geq 0$, where $P^* := P(x^*)$. From the update rule of $w^k$ in Algorithm 1, it is easy to see that

$$\mathbb{E}_k[\mathcal{W}^{k+1}] = (1-p)\mathcal{W}^k + \frac{\theta_2}{q}\mathcal{Y}^k, \tag{10}$$

for $k \geq 0$. Let $B_f(x,y) := f(x) - f(y) - \langle \nabla f(y), x - y \rangle$ be the Bregman divergence and $\mathbb{E}_k[\cdot]$ denote the expectation conditional on $x^k, y^k, z^k, w^k, h_\tau^k, u_\tau^k$, and $e_\tau^k$. The following lemma reveals the evolution of the other two terms $\tilde{\mathcal{Z}}^k$ and $\mathcal{Y}^k$.

**Lemma 3.3.** *If $\mathcal{L}_1 \geq L_f$ and $\theta_1 + \theta_2 \leq 1$, then $\mathbb{E}_k\left[\tilde{\mathcal{Z}}^{k+1} + \mathcal{Y}^{k+1}\right]$ can be upper bounded by*

$$\frac{\mathcal{L}_1}{\mathcal{L}_1 + \eta\mu/2}\tilde{\mathcal{Z}}^k + (1 - \theta_1 - \theta_2)\mathcal{Y}^k + pq\mathcal{W}^k + \left(\frac{\mathcal{L}_1}{2\eta} + \frac{\mu_f}{2}\right)\|e^k\|^2 + \left(\frac{\mathcal{L}_1}{2\eta} + \frac{\mu}{2}\right)\mathbb{E}_k\left[\|e^{k+1}\|^2\right]$$
$$- \frac{1}{\theta_1}\left(\theta_2 - \frac{2L}{n\mathcal{L}_1}\right)B_f(w^k, x^k) - \frac{1 - \theta_1 - \theta_2}{\theta_1}B_f(y^k, x^k).$$

Because of the compression, we have the additional error terms $\|e^k\|^2$ and $\|e^{k+1}\|^2$ in the evolution of $\tilde{\mathcal{Z}}^k$ and $\mathcal{Y}^k$ in Lemma 3.3. To upper bound these error terms, we need to analyze the evolution of $\frac{1}{n}\sum_{\tau=1}^n e_\tau^k$ and $\frac{1}{n}\sum_{\tau=1}^n \|h_\tau^k - \nabla f^{(\tau)}(w^k)\|^2$ in the next two lemmas.

**Lemma 3.4.** *The quantity $\mathbb{E}_k\left[\frac{1}{n}\sum_{\tau=1}^n \|e_\tau^{k+1}\|^2\right]$ is upper bounded by the expression*

$$\left(1 - \frac{\delta}{2}\right)\frac{1}{n}\sum_{\tau=1}^n \|e_\tau^k\|^2 + \frac{4(1-\delta)\eta^2}{\delta n\mathcal{L}_1^2}\sum_{\tau=1}^n \|\nabla f^{(\tau)}(w^k) - h_\tau^k\|^2 + \frac{2(1-\delta)\eta^2}{\mathcal{L}_1^2}\left(\frac{4\bar{L}}{\delta} + L\right)B_f(w^k, x^k).$$

**Lemma 3.5.** *The quantity $\frac{1}{n}\sum_{\tau=1}^n \mathbb{E}_k[\|h_\tau^{k+1} - \nabla f^{(\tau)}(w^{k+1})\|^2]$ is upper bounded by the expression*

$$\left(1 - \frac{\delta_1}{2}\right)\frac{1}{n}\sum_{\tau=1}^n \|h_\tau^k - \nabla f^{(\tau)}(w^k)\|^2 + 4\bar{L}p\left(1 + \frac{2p}{\delta_1}\right)B_f(y^k, x^k) + 4\bar{L}p\left(1 + \frac{2p}{\delta_1}\right)B_f(w^k, x^k).$$

Under the additional Assumption 2.1, we need to analyze the evolution of $\|e^k\|^2$ and $\|h^k - \nabla f(w^k)\|^2$ as well, which can be found in the appendix.

### 3.3 Convergence analysis: main results

With the above lemmas at hand, we are ready to construct suitable Lyapunov functions which enable us to prove linear convergence. First, we construct the Lyapunov function $\Phi^k$ for the general case. Let $\mathcal{L}_2 := \frac{6L}{n} + \frac{112(1-\delta)\bar{L}}{3\delta^2} + \frac{28(1-\delta)L}{3\delta} + \frac{224(1-\delta)\bar{L}p}{\delta^2\delta_1}\left(1 + \frac{2p}{\delta_1}\right)$, and for $k \geq 0$ define

$$\Phi^k := \tilde{\mathcal{Z}}^k + \mathcal{Y}^k + \mathcal{W}^k + \frac{4\mathcal{L}_1}{\delta\eta} \cdot \frac{1}{n}\sum_{\tau=1}^n \|e_\tau^k\|^2 + \frac{56(1-\delta)}{3\theta_1\delta^2\delta_1\mathcal{L}_1} \cdot \frac{1}{n}\sum_{\tau=1}^n \|h_\tau^k - \nabla f^{(\tau)}(w^k)\|^2.$$

Then we can get the linear convergence of $\Phi^k$ in the following theorem.

**Theorem 3.6.** *Let Assumption 3.1 and Assumption 3.2 hold. If $\mathcal{L}_1 \geq \max\{L_f, 3\mu\eta\}$, $\theta_1 + 2\theta_2 \leq 1$, and $\theta_2 \geq \frac{\mathcal{L}_2}{3\mathcal{L}_1}$, then we have*

$$\mathbb{E}\left[\Phi^k\right] \leq \left(1 - \min\left(\frac{\mu}{\mu+6\theta_1\mathcal{L}_1}, \theta_1 + \theta_2 - \frac{\theta_2}{q}, p(1-q), \frac{\delta}{6}, \frac{\delta_1}{6}\right)\right)^k \Phi^0, \quad \forall k \geq 0.$$

If Assumption 2.1 holds, we define the Lyapunov function $\Psi^k$ as follows. Let $\mathcal{L}_3 := \frac{6L}{n} + \frac{112(1-\delta)L_f}{3\delta^2} + \frac{224(1-\delta)\bar{L}}{\delta^2 n} + \frac{308(1-\delta)L}{3\delta n} + \frac{224(1-\delta)p}{\delta^2\delta_1}\left(L_f + \frac{9\bar{L}}{n}\right)\left(1 + \frac{2p}{\delta_1}\right)$, and for $k \geq 0$ define

$$\begin{aligned}
\Psi^k \quad := \quad &\tilde{\mathcal{Z}}^k + \mathcal{Y}^k + \mathcal{W}^k + \frac{4\mathcal{L}_1}{\delta\eta}\|e^k\|^2 + \frac{28\mathcal{L}_1(1-\delta)}{\delta\eta n} \cdot \frac{1}{n}\sum_{\tau=1}^n \|e_\tau^k\|^2 \\
&+ \frac{56(1-\delta)\eta}{\delta^2\delta_1\mathcal{L}_1}\|h^k - \nabla f(w^k)\|^2 + \frac{504(1-\delta)\eta}{\delta^2\delta_1 n\mathcal{L}_1} \cdot \frac{1}{n}\sum_{\tau=1}^n \|h_\tau^k - \nabla f^{(\tau)}(w^k)\|^2.
\end{aligned}$$

Then we can get the linear convergence of $\Psi^k$ in Theorem 3.7.

**Theorem 3.7.** *Let Assumption 2.1 , Assumption 3.1 and Assumption 3.2 hold. If $\mathcal{L}_1 \geq \max\{L_f, 3\mu\eta\}$, $\theta_1 + 2\theta_2 \leq 1$, and $\theta_2 \geq \frac{\mathcal{L}_3}{3\mathcal{L}_1}$, then we have*

$$\mathbb{E}\left[\Psi^k\right] \leq \left(1 - \min\left(\frac{\mu}{\mu+6\theta_1\mathcal{L}_1}, \theta_1 + \theta_2 - \frac{\theta_2}{q}, p(1-q), \frac{\delta}{6}, \frac{\delta_1}{6}\right)\right)^k \Psi^0, \quad \forall k \geq 0.$$

From Theorems 3.6 and 3.7, by choosing the parameters appropriately, we can obtain the iteration complexity for ECLK.

**Theorem 3.8.** *Let Assumption 3.1 and Assumption 3.2 hold. Let $\mathcal{L}_1 = \max(\mathcal{L}_4, L_f, 3\mu\eta)$, $\theta_2 = \frac{\mathcal{L}_4}{3\max\{L_f, \mathcal{L}_4\}}$, and*

$$\theta_1 = \begin{cases} \min\left(\sqrt{\frac{\mu}{\mathcal{L}_4 p}}\theta_2, \theta_2\right) & \text{if } L_f \leq \frac{\mathcal{L}_4}{p} \\ \min\left(\sqrt{\frac{\mu}{L_f}}, \frac{p}{3}\right) & \text{otherwise} \end{cases}.$$

*(i) Let $\mathcal{L}_4 = \mathcal{L}_2$. Then with some $q \in [\frac{2}{3}, 1)$, we have $\mathbb{E}[\Phi^k] \leq \epsilon\Phi^0$ for*

$$k \geq \mathcal{O}\left(\left(\frac{1}{\delta} + \frac{1}{\delta_1} + \frac{1}{p} + \sqrt{\frac{L_f}{\mu}} + \sqrt{\frac{\mathcal{L}_2}{\mu p}}\right)\log\frac{1}{\epsilon}\right).$$

*In particular, if $p \leq \mathcal{O}(\delta_1)$, then the iteration complexity becomes*

$$k \geq \mathcal{O}\left(\left(\frac{1}{\delta} + \frac{1}{p} + \sqrt{\frac{L_f}{\mu}} + \sqrt{\frac{L}{\mu pn}} + \frac{1}{\delta}\sqrt{\frac{(1-\delta)\bar{L}}{\mu p}} + \sqrt{\frac{(1-\delta)L}{\mu p\delta}}\right)\log\frac{1}{\epsilon}\right). \tag{11}$$

*(ii) Let $\mathcal{L}_4 = \mathcal{L}_3$. If Assumption 2.1 holds, then for some $q \in [\frac{2}{3}, 1)$, we have $\mathbb{E}[\Psi^k] \leq \epsilon\Psi^0$ for $k \geq \mathcal{O}\left(\left(\frac{1}{\delta} + \frac{1}{\delta_1} + \frac{1}{p} + \sqrt{\frac{L_f}{\mu}} + \sqrt{\frac{\mathcal{L}_3}{\mu p}}\right)\log\frac{1}{\epsilon}\right)$. If $p \leq \mathcal{O}(\delta_1)$, the iteration complexity becomes*

$$k \geq \mathcal{O}\left(\left(\frac{1}{\delta} + \frac{1}{p} + \sqrt{\frac{L_f}{\mu}} + \sqrt{\frac{L}{\mu pn}} + \frac{1}{\delta}\sqrt{\frac{(1-\delta)L_f}{\mu p}} + \sqrt{\frac{(1-\delta)L}{\mu p\delta n}}\right)\log\frac{1}{\epsilon}\right). \tag{12}$$

From Theorem 3.8, it is easy to see that the optimal $p$ for ECLK is $\Theta(\delta_1)$.

**Comparison to ADIANA.** ADIANA [Li et al., 2020] is an accelerated and compressed distributed method with any unbiased compressor. The iteration complexity of ADIANA is $\mathcal{O}\left(\omega\left(1+\sqrt{\frac{\bar{L}}{n\mu}}\right)\log\frac{1}{\epsilon}\right)$ when $n \leq \omega$, and $\mathcal{O}\left(\left(\omega+\sqrt{\frac{\bar{L}}{\mu}}+\sqrt{\sqrt{\frac{\omega}{n}}\frac{\omega\bar{L}}{\mu}}\right)\log\frac{1}{\epsilon}\right)$ when $n > \omega$. If we choose $Q$ and $Q_1$ in ECLK to be the contraction compressor $\frac{1}{\omega+1}\tilde{Q}$ where $\tilde{Q}$ is any unbiased compressor, then $\delta = \delta_1 = \frac{1}{\omega+1}$, and thus the complexity in (12) becomes

$$\mathcal{O}\left(\left(\omega+\sqrt{\frac{L_f}{\mu}}+\sqrt{\frac{(\omega+1)L}{\mu n}}+(\omega+1)\sqrt{\frac{\omega L_f}{\mu}}+\sqrt{\frac{\omega(\omega+1)L}{\mu n}}\right)\log\frac{1}{\epsilon}\right),$$

for $p = \delta_1$. In this case, the dependency of $\omega$ of the iteration complexity of ADIANA is better than that of ECLK. However, the full gradients are computed by all nodes at each iteration in ADIANA, which could be slower than the communication when $m$ is very large. Furthermore, the contraction compressor could be more efficient than the unbiased compressor in practice, i.e., for the same level of compression, $\delta$ could be larger than $1/(\omega+1)$. This efficiency of the contraction compressor is the motivation to study the error compensated methods, and in numerical experiments, ECLK is comparable to ADIANA.

## 4 Experiments

In this section, we experimentally study the performance of error compensated L-Katyusha (ECLK) used with several contraction compressors on the logistic regression problem for binary classification,

$$x \mapsto \log\left(1+\exp(-y_i A_i^T x)\right) + \frac{\lambda}{2}\|x\|^2,$$

where $\{A_i, y_i\}$ are the training data points. We use Python 3.7 to perform the experiments. We use the datasets `a5a`, `a9a`, `w6a`, `w8a`, `phishing`, and `mushrooms` from the LIBSVM library [Chang and Lin, 2011]. The regularization parameter was set to $\lambda = 10^{-3}$. The number of nodes in our experiments is $n = 20$, and the optimal solution is obtained by running the uncompressed L-Katyusha for $10^5$ iterations. More experiments can be found in the appendix.

**Compressors.** We use three contraction compressors: the TopK compressor with $K = 1$, or the compressor $\frac{1}{\omega+1}\tilde{Q}$, where $\tilde{Q}$ is either the unbiased random dithering compressor with $s = \sqrt{d}$ levels [Alistarh et al., 2017] or the natural compressor [Horváth et al., 2019b]. For the Top1 compressor, the number of communicated bits for the compressed vector is $64 + \lceil\log d\rceil$, and $\delta = 1/d$. For the random dithering compressor with $s = \sqrt{d}$, the number of communicated bits for the compressed vector is $2.8d + 64$, and $\omega = 1$ [Alistarh et al., 2017] . For the natural compressor, the number of communicated bits for the compressed vector is $12d$, and $\omega = 1/8$ [Horváth et al., 2019b].

**Parameter setting.** In all the experiments, we search for the optimal stepsize for all tested algorithms. In particular, we use the parameter setting in Theorem 3.8 (i) for ECLK. From Theorem 3.8 (i), we know that we can set values for all parameters if we know values of $\mu$, $\delta$, $\delta_1$, $p$, and Lipschitz constants. For the tested problem, $\mu = \lambda$. $Q$ and $Q_1$ are chosen to be the same contraction compressor for ECLK, which implies that $\delta = \delta_1$, and the lower bound for $\delta$ can be obtained easily for our used compressors. We set $p = \delta$ for ECLK except in Section 4.4. We calculate the theoretical $L_f$, $\bar{L}$, and $L$ as $L_f^{th}$, $\bar{L}^{th}$, and $L^{th}$, respectively. Then we choose $L_f = t \cdot L_f^{th}$, $\bar{L} = t \cdot \bar{L}^{th}$, and $L = t \cdot L^{th}$, and search for the best $t$ in the set $t \in \{10^{-k} \mid k = 0, 1, 2, ...\}$. We set $p = \delta$ for EC-LSVRG-DIANA [Gorbunov et al., 2020]. EC-LSVRG-DIANA requires the use of an additional unbiased compressor; for this we make use of random dithering.

### 4.1 Top1 vs random dithering vs natural compression vs no compression

First, we compare the uncompressed L-Katyusha method with ECLK with three contraction compressors: Top1, random dithering, and natural compression; see Figure 1. ECLK is orders of magnitude better in terms of communication complexity.

### 4.2 Comparison with EC-LSVRG-DIANA and ECGD

Next, we compare ECLK with EC-LSVRG-DIANA [Gorbunov et al., 2020] and error compensated GD (ECGD) using the Top1, random dithering, and natural compressor. Note that ECGD is a special

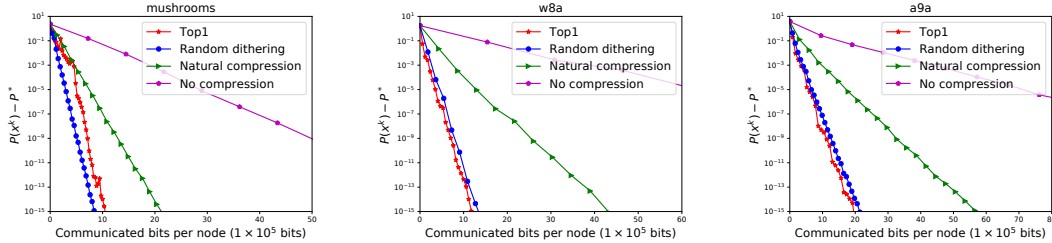

Figure 1: The communication complexity performance of ECLK used with compressors: Top1 vs random dithering vs natural compression vs no compression. Datasets: `mushrooms`, `w8a`, and `a9a`.

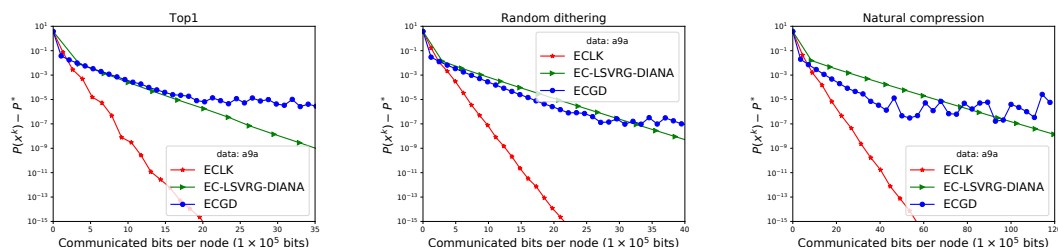

Figure 2: The communication complexity performance of ECGD vs EC-LSVRG-DIANA vs ECLK for Top1, Random dithering, and Natural compression on `a9a` data set.

case of error compensated SGD [Stich et al., 2018] with $m = 1$, where the full gradient $\nabla f^{(\tau)}(x^k)$ is calculated on each node. Figures 2 and 3 show that ECGD can only converge to a neighborhood of the optimal solution, and that ECLK is considerably faster than EC-LSVRG-DIANA.

### 4.3 Comparison with ADIANA

Let us denote by ECLK-F the special case of ECLK with $m = 1$, i.e., the full gradient $\nabla f^{(\tau)}(x^k)$ is calculated on each node. We compare ECLK-F with the accelerated variant of DIANA, called ADIANA [Li et al., 2020], for six data sets. For ADIANA we use two unbiased compressors: random dithering and natural compression. Figure 4 shows that in terms of communication complexity, ECLK-F is comparable to ADIANA, and can be better than ADIANA in some cases.

### 4.4 Impact of the update frequency of the reference point

Finally, we test the impact of the update frequency of the reference point $p$ for ECLK with Top1 compressor; see Figure 5. We choose five values for $p$: $\delta/3$, $\delta$, $3\delta$, $9\delta$, and 1. Figure 5 shows that the performance of $p = \delta$ is usually better than $p = \delta/3$. As we increase $p$ from $p = \delta$, the performance of ECLK could be improved. However, if $p$ is too large, the performance of ECLK is no better than $p = \delta$, generally.

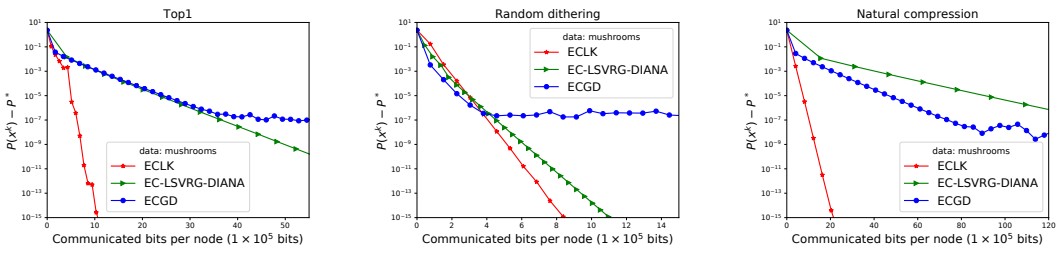

Figure 3: The communication complexity performance of ECGD vs EC-LSVRG-DIANA vs ECLK for Top1, Random dithering, and Natural compression on `mushrooms` data set.

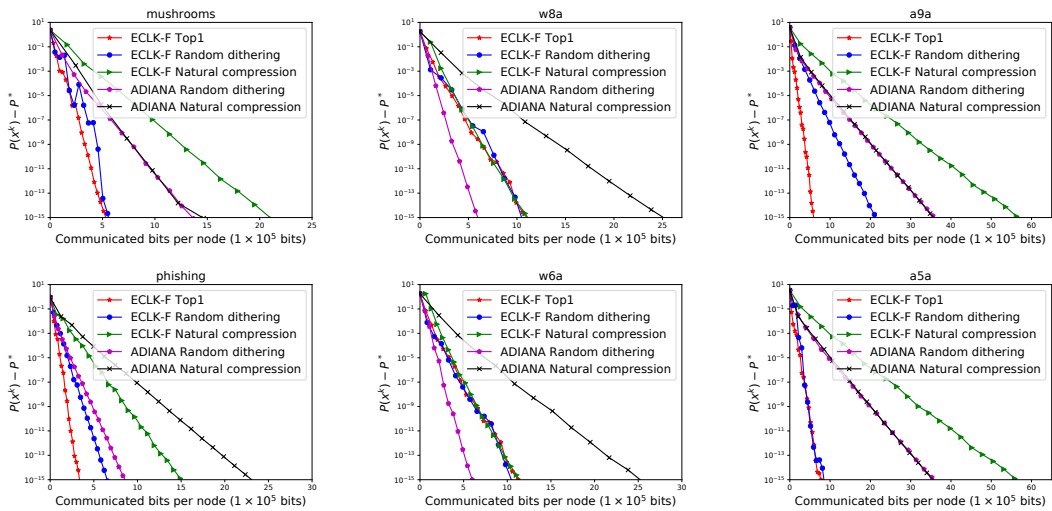

Figure 4: The communication complexity performance of ECLK vs ADIANA for Top1, Random dithering, and Natural compression on `mushrooms`, `w8a`, `a9a`, `phishing`, `w6a`, and `a5a` data sets.

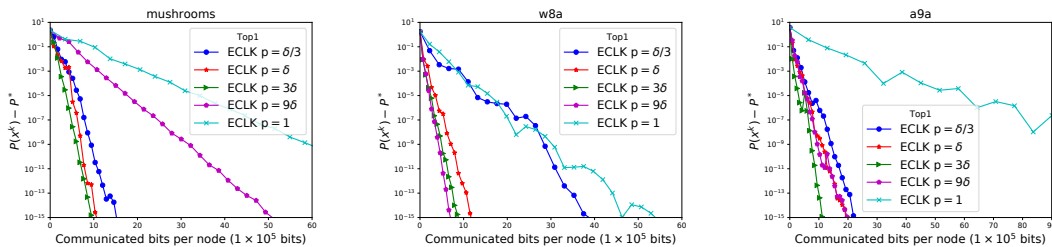

Figure 5: The communication complexity performance of ECLK with the Top1 compressor and $p \in \{\delta/3, \delta, 3\delta, 9\delta, 1\}$ on the `mushrooms`, `w8a`, and `a9a` data sets.

## Acknowledgments and Disclosure of Funding

Xun Qian and Peter Richtárik acknowledge funding by the KAUST Baseline Research Funding Scheme, the Extreme Computing Research Center at KAUST, and administrative support from the Visual Computing Center at KAUST. Tong Zhang acknowledges further funding by GRF 16201320.

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
