in Figure 6 for `phishing`, `w6a`, and `a5a` data sets. ECLK is much better in terms of communication complexity.

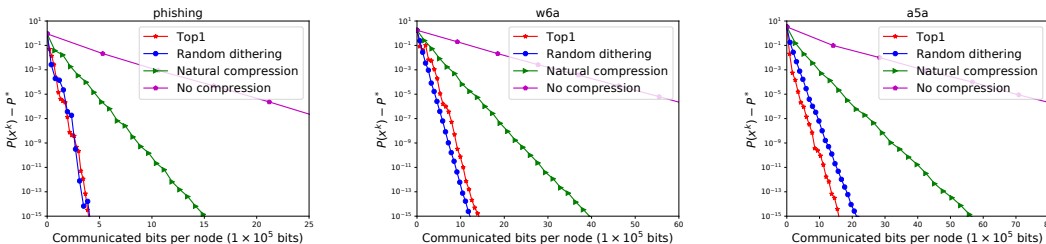

Figure 6: The communication complexity performance of ECLK used with compressors: Top1 vs random dithering vs natural compression vs no compression. Datasets: `phishing`, `w6a`, and `a5a`.

**Comparison with EC-LSVRG-DIANA and ECGD.** We compare ECLK with EC-LSVRG-DIANA and ECGD using the Top1, random dithering, and natural compression for `w8a`, `phishing`, `w6a`, and `a5a` data sets in Figures 7, 8, 9, and 10. These figures show that ECGD can only converge to a neighborhood of the optimal solution, and that ECLK is usually better than EC-LSVRG-DIANA in terms of communication complexity.

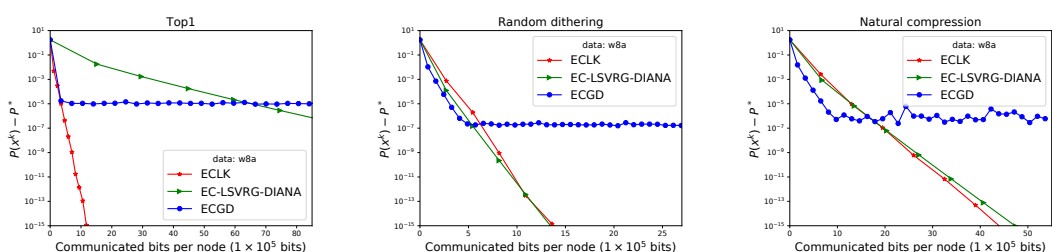

Figure 7: The communication complexity performance of ECGD vs EC-LSVRG-DIANA vs ECLK for Top1, Random dithering, and Natural compression on `w8a` data set.

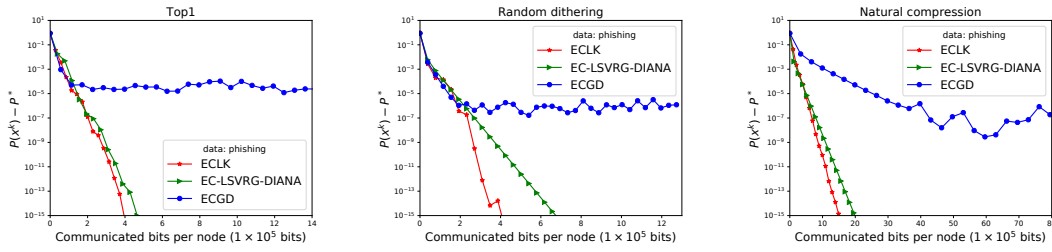

Figure 8: The communication complexity performance of ECGD vs EC-LSVRG-DIANA vs ECLK for Top1, Random dithering, and Natural compression on `phishing` data set.

**ECLK vs ADIANA.** We compared ECLK-F with ADIANA in Figure 4. Now we also compare ECLK with ADIANA. Figure 11 shows that even though in ECLK we calculate the stochastic gradient and in ADIANA we compute the full gradient, ECLK is comparable to ADIANA, and can be better than ADIANA in some cases as well.

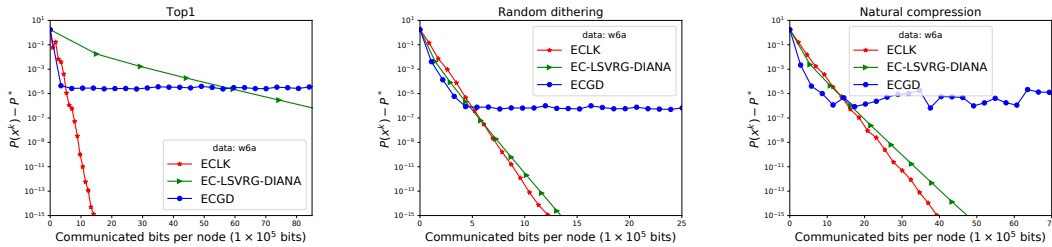

Figure 9: The communication complexity performance of ECGD vs EC-LSVRG-DIANA vs ECLK for Top1, Random dithering, and Natural compression on `w6a` data set.

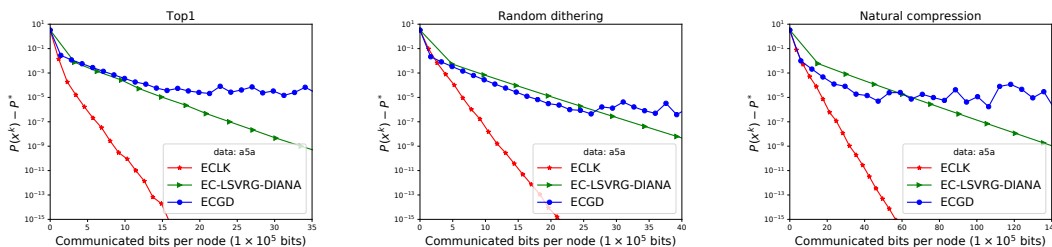

Figure 10: The communication complexity performance of ECGD vs EC-LSVRG-DIANA vs ECLK for Top1, Random dithering, and Natural compression on `a5a` data set.

**Impact of the update frequency of the reference point.** We test the impact of the update frequency of the reference point $p$ for ECLK with Top1 compressor on `phishing`, `w6a`, and `a5a` data sets in Figure 12. We choose five values for $p$: $\delta/3$, $\delta$, $3\delta$, $9\delta$, and $1$. Similar conclusions can be made as in subsection 4.4.

We test the impact of $p$ for ECLK with random dithering compressor in Figure 13. Noticing that $p = \delta = 1/2$ in this case, we choose five values for $p$: $\delta/27$, $\delta/9$, $\delta/3$, $\delta$, and $1$. Figure 13 shows that we may get better performance by decreasing $p$ from $\delta$. However, if $p$ is too large or too small, the performance of ECLK is generally no better than $p = \delta$.

We test the impact of $p$ for ECLK with natural compression in Figure 14. Noticing that $p = \delta = 8/9$ in this case, we choose five values for $p$: $\delta/27$, $\delta/9$, $\delta/3$, $\delta$, and $1$. The performace of ECLK with $p = 1$ is colse to the $p = \delta$ case. Sometimes we may get better performance by decreasing $p$ from $\delta$. However, if $p$ is too small, the performance of ECLK is no better than $p = \delta$, generally.

**ECLK vs ECLK-F vs EC-LSVRG-DIANA vs ADIANA with $\lambda = 10^{-5}$.** We compare ECLK with ECLK-F, EC-LSVRG-DIANA, and ADIANA for Top1 and Random dithering (RD) with $\lambda = 10^{-5}$ in Figure 15. Figure 15 shows that ECLK-F with Top1 compressor has the best performace in terms of communication complexity, and EC-LSVRG-DIANA is usually much slower than these accelerated algorithms : ECLK, ECLK-F, and ADIANA.

**ECLK vs EC-LSVRG-DIANA for Top1 compressor.** Noticed that for the unbiased compressor in EC-LSVRG-DIANA, we used the random dithering compressor. However, if we use Top1 compressor for the contraction compressor in EC-LSVRG-DIANA, the communication cost of the compressed vector using random dithering is much higher than that using Top1. Let EC-LSVRG-DIANA-2 denote EC-LSVRG-DIANA where we use Rand1 for the unbiased compressor (RandK can be transformed to an unbiased compressor by scaling). We compare ECLK with EC-LSVRG-DIANA and EC-LSVRG-DIANA-2 for Top1 with $\lambda = 10^{-3}$ and $\lambda = 10^{-5}$ in Figure 16 and Figure 17, respectively. Figure 16 and Figure 17 show that except for the case where `phishing` and $\lambda = 10^{-3}$ are used, EC-LSVRG-DIANA-2 is better than EC-LSVRG-DIANA, and that ECLK is generally much better than EC-LSVRG-DIANA-2, especially for $\lambda = 10^{-5}$.

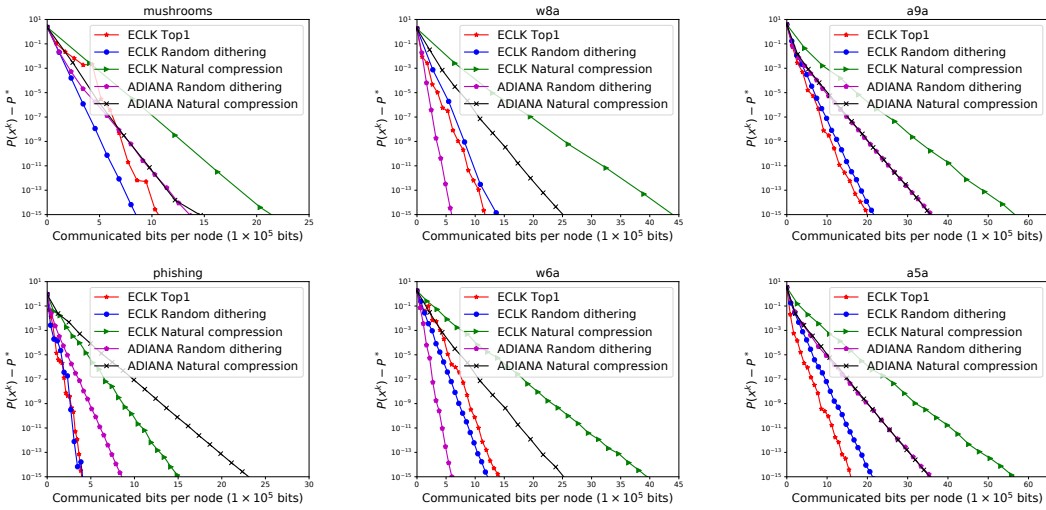

Figure 11: The communication complexity performance of ECLK vs ADIANA for Top1, Random dithering, and Natural compression on `mushrooms`, `w8a`, `a9a`, `phishing`, `w6a`, and `a5a` data sets.

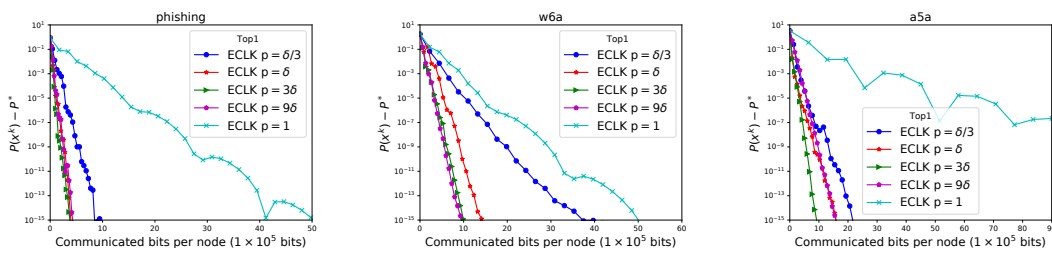

Figure 12: The communication complexity performance of ECLK with the Top1 compressor and $p \in \{\delta/3, \delta, 3\delta, 9\delta, 1\}$ on the `phishing`, `w6a`, and `a5a` data sets.

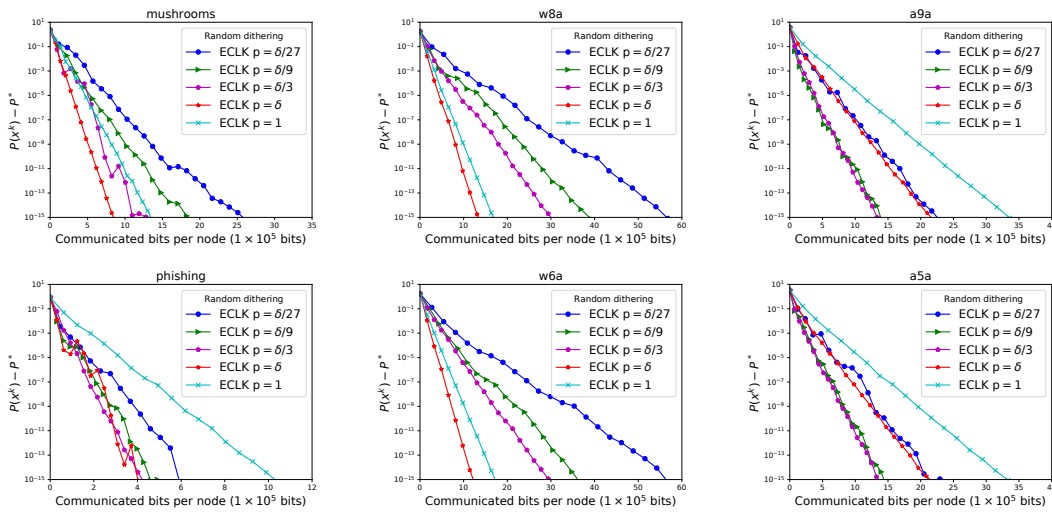

Figure 13: The communication complexity performance of ECLK with the random dithering compressor and $p \in \{\delta/27, \delta/9, \delta/3, \delta, 1\}$ on the `mushrooms`, `w8a`, `a9a`, `phishing`, `w6a`, and `a5a` data sets.

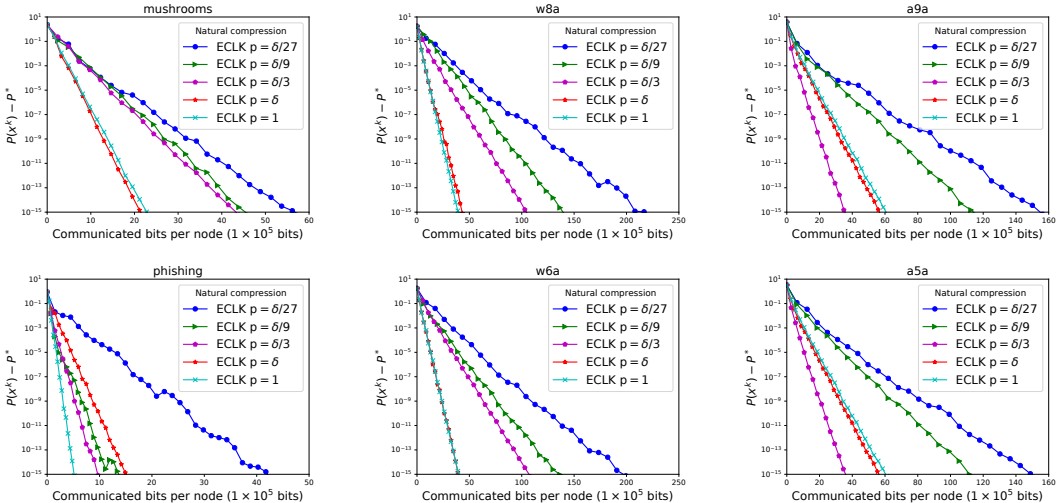

Figure 14: The communication complexity performance of ECLK with the natural compression compressor and $p \in \{\delta/27, \delta/9, \delta/3, \delta, 1\}$ on the `mushrooms`, `w8a`, `a9a`, `phishing`, `w6a`, and `a5a` data sets.

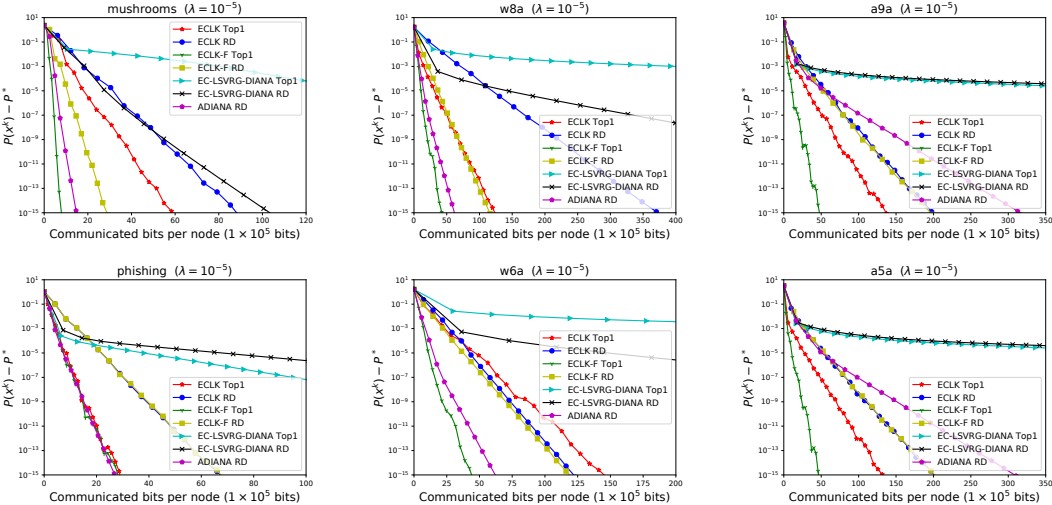

Figure 15: The communication complexity performance of ECLK vs ECLK-F vs EC-LSVRG-DIANA vs ADIANA for Top1 and Random dithering (RD) with $\lambda = 10^{-5}$ on `mushrooms`, `w8a`, `a9a`, `phishing`, `w6a`, and `a5a` data sets.

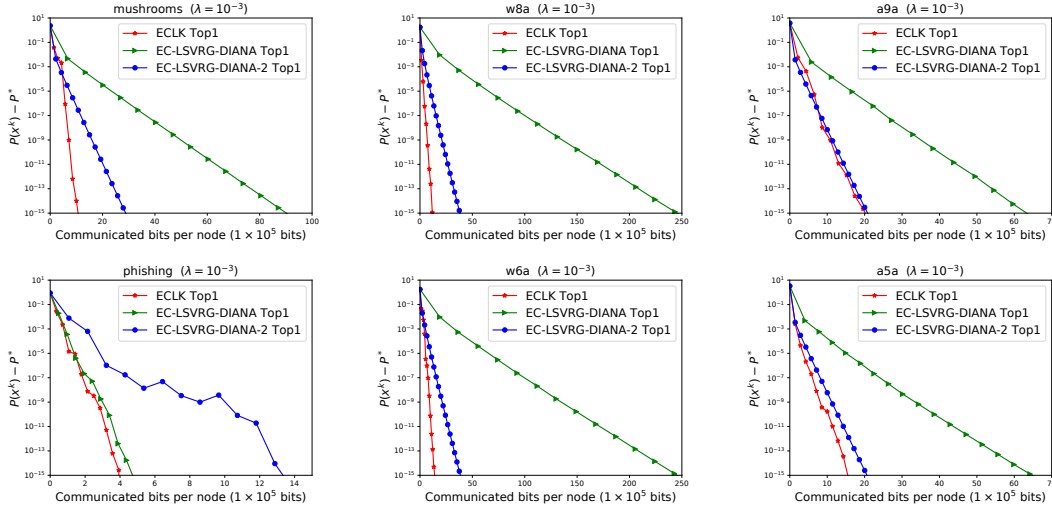

Figure 16: The communication complexity performance of ECLK vs EC-LSVRG-DIANA vs EC-LSVRG-DIANA-2 for Top1 with $\lambda = 10^{-3}$ on mushrooms, w8a, a9a, phishing, w6a, and a5a data sets.

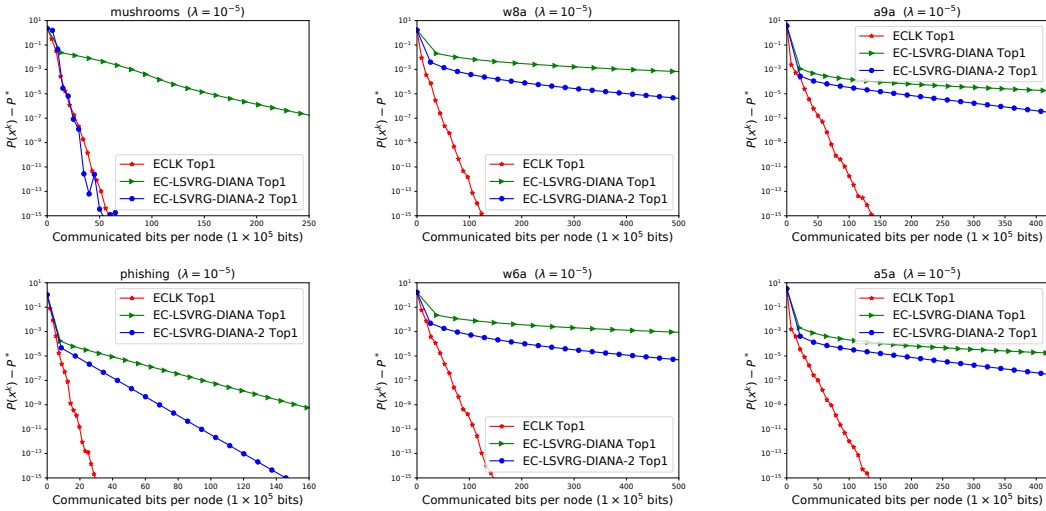

Figure 17: The communication complexity performance of ECLK vs EC-LSVRG-DIANA vs EC-LSVRG-DIANA-2 for Top1 with $\lambda = 10^{-5}$ on mushrooms, w8a, a9a, phishing, w6a, and a5a data sets.

# B  Lemmas

**Lemma B.1.** *We have*

$$\frac{1}{n}\sum_{\tau=1}^{n}\mathbb{E}_k\left\|\nabla f_{i_k^\tau}^{(\tau)}(x^k)-\nabla f_{i_k^\tau}^{(\tau)}(w^k)\right\|^2 \leq 2L\left(f(w^k)-f(x^k)-\langle\nabla f(x^k),w^k-x^k\rangle\right), \quad (13)$$

*and*

$$\frac{1}{n}\sum_{\tau=1}^{n}\left\|\nabla f^{(\tau)}(x^k)-\nabla f^{(\tau)}(w^k)\right\|^2 \leq 2\bar{L}\left(f(w^k)-f(x^k)-\langle\nabla f(x^k),w^k-x^k\rangle\right), \quad (14)$$

*and*

$$\mathbb{E}_k\left\|\frac{1}{n}\sum_{\tau=1}^{n}\left(\nabla f_{i_k^\tau}^{(\tau)}(x^k)-\nabla f_{i_k^\tau}^{(\tau)}(w^k)\right)\right\|^2$$
$$\leq \quad 2\left(L_f+\frac{L}{n}\right)\left(f(w^k)-f(x^k)-\langle\nabla f(x^k),w^k-x^k\rangle\right), \quad (15)$$

*and*

$$\mathbb{E}_k\left[\|g^k+h^k-\nabla f(x^k)\|^2\right] \quad \leq \quad \frac{2L}{n}\left(f(w^k)-f(x^k)-\langle\nabla f(x^k),w^k-x^k\rangle\right), \quad (16)$$

Next two lemmas will be used to prove Lemma 3.3.

**Lemma B.2.** *If $\mathcal{L}_1 \geq L_f$, then we have*

$$\frac{\mathcal{L}_1}{4\eta}\|z^{k+1}-z^k\|^2+\langle g^k+h^k,z^{k+1}-z^k\rangle \geq \frac{1}{\theta_1}\left(f(y^{k+1})-f(x^k)\right)-\frac{1}{\mathcal{L}_1\theta_1}\|g^k+h^k-\nabla f(x^k)\|^2. \quad (17)$$

**Lemma B.3.** *We have*

$$\langle g^k+h^k,x^*-z^{k+1}\rangle+\frac{\mu_f}{2}\|x^k-x^*\|^2$$
$$\geq \quad \frac{\mathcal{L}_1}{4\eta}\|z^k-z^{k+1}\|^2+\tilde{\mathcal{Z}}^{k+1}-\frac{\mathcal{L}_1\tilde{\mathcal{Z}}^k}{\mathcal{L}_1+\eta\mu/2}-\left(\frac{\mathcal{L}_1}{2\eta}+\frac{\mu_f}{2}\right)\|e^k\|^2$$
$$-\left(\frac{\mathcal{L}_1}{2\eta}+\frac{\mu}{2}\right)\|e^{k+1}\|^2+\psi(z^{k+1})-\psi(x^*). \quad (18)$$

Under the additional Assumption 2.1, we analyze the evolution of $\|e^k\|^2$ and $\|h^k-\nabla f(w^k)\|^2$ in the next two lemmas.

**Lemma B.4.** *Under Assumption 2.1, the quantity $\mathbb{E}_k[\|e^{k+1}\|^2]$ is upper bounded by*

$$\left(1-\frac{\delta}{2}\right)\|e^k\|^2+\frac{2(1-\delta)\delta}{n^2}\sum_{\tau=1}^{n}\|e_\tau^k\|^2+\frac{4(1-\delta)\delta\eta^2}{n^2\mathcal{L}_1^2}\sum_{\tau=1}^{n}\|\nabla f^{(\tau)}(w^k)-h_\tau^k\|^2$$
$$+\frac{2(1-\delta)\eta^2}{\mathcal{L}_1^2}\left(\frac{4L_f}{\delta}+\frac{5L}{n}\right)B_f(w^k,x^k)+\frac{4(1-\delta)\eta^2}{\delta\mathcal{L}_1^2}\|h^k-\nabla f(w^k)\|^2.$$

**Lemma B.5.** *Under Assumption 2.1, the quantity $\mathbb{E}_k[\|h^{k+1}-\nabla f(w^{k+1})\|^2]$ is upper bounded by*

$$\left(1-\frac{\delta_1}{2}\right)\|h^k-\nabla f(w^k)\|^2+\frac{(1-\frac{\delta_1}{2})\delta_1}{n^2}\sum_{\tau=1}^{n}\|h_\tau^k-\nabla f^{(\tau)}(w^k)\|^2$$
$$+4pL_f\left(1+\frac{2p}{\delta_1}\right)B_f(y^k,x^k)+4pL_f\left(1+\frac{2p}{\delta_1}\right)B_f(w^k,x^k).$$

## C Proofs of Lemma B.1, Lemma B.2, Lemma B.3, and Lemma 3.3

### C.1 Proof of Lemma B.1

Since $f_i^{(\tau)}$ is $L$-smooth and $f$ is $L_f$-smooth, we have (Nesterov [2004], Theorem 2.1.5)

$$\|\nabla f_i^{(\tau)}(x) - \nabla f_i^{(\tau)}(y)\|^2 \le 2L(f_i^{(\tau)}(x) - f_i^{(\tau)}(y) - \langle \nabla f_i^{(\tau)}(y), x - y \rangle),$$

for any $x, y \in \mathbb{R}^d$. Therefore,

$$
\begin{aligned}
\mathbb{E}_k \|\nabla f_{i_k^\tau}^{(\tau)}(x^k) - \nabla f_{i_k^\tau}^{(\tau)}(w^k)\|^2 &\le 2L\mathbb{E}_k[f_{i_k^\tau}^{(\tau)}(w^k) - f_{i_k^\tau}^{(\tau)}(x^k) - \langle \nabla f_{i_k^\tau}^{(\tau)}(x^k), w^k - x^k \rangle] \\
&= 2L(f^{(\tau)}(w^k) - f^{(\tau)}(x^k) - \langle \nabla f^{(\tau)}(x^k), w^k - x^k \rangle),
\end{aligned}
$$

which implies that (13). From

$$\|\nabla f^{(\tau)}(x) - \nabla f^{(\tau)}(y)\|^2 \le 2\bar{L}\left(f^{(\tau)}(x) - f^{(\tau)}(y) - \langle \nabla f^{(\tau)}(y), x - y \rangle\right),$$

we can prove (14) similarly.

Denote $q_\tau^k = \nabla f_{i_k^\tau}^{(\tau)}(x^k) - \nabla f_{i_k^\tau}^{(\tau)}(w^k)$. Then we have

$$
\begin{aligned}
&\mathbb{E}_k \left\|\frac{1}{n}\sum_{\tau=1}^n \left(\nabla f_{i_k^\tau}^{(\tau)}(x^k) - \nabla f_{i_k^\tau}^{(\tau)}(w^k)\right)\right\|^2 \\
=\ & \mathbb{E}_k \left\|\frac{1}{n}\sum_{\tau=1}^n q_\tau^k\right\|^2 \\
=\ & \frac{1}{n^2}\mathbb{E}_k \left\langle \sum_{\tau=1}^n q_\tau^k, \sum_{\tau=1}^n q_\tau^k \right\rangle \\
=\ & \frac{1}{n^2}\sum_{\tau_1,\tau_2=1}^n \mathbb{E}_k \left\langle q_{\tau_1}^k, q_{\tau_2}^k \right\rangle \\
=\ & \frac{1}{n^2}\sum_{\tau=1}^n \mathbb{E}_k \|q_\tau^k\|^2 + \frac{1}{n^2}\sum_{\tau_1 \ne \tau_2} \left\langle \nabla f^{(\tau_1)}(x^k) - \nabla f^{(\tau_1)}(w^k), \nabla f^{(\tau_2)}(x^k) - \nabla f^{(\tau_2)}(w^k) \right\rangle \\
=\ & \frac{1}{n^2}\sum_{\tau=1}^n \mathbb{E}_k \|q_\tau^k\|^2 + \|\nabla f(x^k) - \nabla f(w^k)\|^2 - \frac{1}{n^2}\sum_{\tau=1}^n \|\nabla f^{(\tau)}(x^k) - \nabla f^{(\tau)}(w^k)\|^2 \quad (19) \\
\le\ & \frac{1}{n^2}\sum_{\tau=1}^n \mathbb{E}_k \|q_\tau^k\|^2 + 2L_f(f(w^k) - f(x^k) - \langle \nabla f(x^k), w^k - x^k \rangle) \\
\overset{(13)}{\le}\ & \left(\frac{2L}{n} + 2L_f\right)(f(w^k) - f(x^k) - \langle \nabla f(x^k), w^k - x^k \rangle),
\end{aligned}
$$

where we use the independence of $i_k^\tau$ in the fourth equality and $\|\nabla f(x) - \nabla f(y)\|^2 \le 2L_f(f(x) - f(y) - \langle \nabla f(y), x - y \rangle)$ in the first inequality.

Recall that $q_\tau^k = \nabla f_{i_k^\tau}^{(\tau)}(x^k) - \nabla f_{i_k^\tau}^{(\tau)}(w^k)$. Then $g^k + h^k = \frac{1}{n}\sum_{\tau=1}^n q_\tau^k + \nabla f(w^k)$, and thus

$$
\begin{aligned}
\mathbb{E}_k\left[\|g^k + h^k - \nabla f(x^k)\|^2\right] &= \mathbb{E}_k\left[\left\|\frac{1}{n}\sum_{\tau=1}^n q_\tau^k + \nabla f(w^k) - \nabla f(x^k)\right\|^2\right] \\
&= \mathbb{E}_k\left\|\frac{1}{n}\sum_{\tau=1}^n q_\tau^k\right\|^2 - \|\nabla f(x^k) - \nabla f(w^k)\|^2 \\
&\overset{(19)}{=} \frac{1}{n^2}\sum_{\tau=1}^n \mathbb{E}_k\|q_\tau^k\|^2 - \frac{1}{n^2}\sum_{\tau=1}^n \mathbb{E}\|\nabla f^{(\tau)}(x^k) - \nabla f^{(\tau)}(w^k)\|^2 \\
&\leq \frac{1}{n^2}\sum_{\tau=1}^n \mathbb{E}_k\|q_\tau^k\|^2 \\
&\leq \frac{2L}{n^2}\sum_{\tau=1}^n \mathbb{E}_k\left(f_{i_k^\tau}^{(\tau)}(w^k) - f_{i_k^\tau}^{(\tau)}(x^k) - \langle\nabla f_{i_k^\tau}^{(\tau)}(x^k), w^k - x^k\rangle\right) \\
&= \frac{2L}{n}\left(f(w^k) - f(x^k) - \langle\nabla f(x^k), w^k - x^k\rangle\right),
\end{aligned}
$$

where we use $\mathbb{E}_k[\frac{1}{n}\sum_{\tau=1}^n q_\tau^k] = \nabla f(x^k) - \nabla f(w^k)$ in the second equality.

## C.2 Proof of Lemma B.2

Since $z^{k+1} - z^k = \frac{1}{\theta_1}(y^{k+1} - x^k)$ and $\eta = \frac{1}{3\theta_1}$, we have

$$
\begin{aligned}
\frac{\mathcal{L}_1}{4\eta}\|z^{k+1} - z^k\|^2 + \langle g^k + h^k, z^{k+1} - z^k\rangle &= \frac{\mathcal{L}_1}{4\eta\theta_1^2}\|y^{k+1} - x^k\|^2 + \frac{1}{\theta_1}\langle g^k + h^k, y^{k+1} - x^k\rangle \\
&= \frac{1}{\theta_1}\langle\nabla f(x^k), y^{k+1} - x^k\rangle + \frac{3\mathcal{L}_1}{4\theta_1}\|y^{k+1} - x^k\|^2 \\
&\quad + \frac{1}{\theta_1}\langle g^k + h^k - \nabla f(x^k), y^{k+1} - x^k\rangle \\
&\geq \frac{1}{\theta_1}\left(f(y^{k+1}) - f(x^k)\right) + \left(\frac{3\mathcal{L}_1}{4\theta_1} - \frac{L_f}{2\theta_1}\right)\|y^{k+1} - x^k\|^2 \\
&\quad + \frac{1}{\theta_1}\langle g^k + h^k - \nabla f(x^k), y^{k+1} - x^k\rangle \\
&\geq \frac{1}{\theta_1}\left(f(y^{k+1}) - f(x^k)\right) + \frac{\mathcal{L}_1}{4\theta_1}\|y^{k+1} - x^k\|^2 \\
&\quad + \frac{1}{\theta_1}\langle g^k + h^k - \nabla f(x^k), y^{k+1} - x^k\rangle \\
&\geq \frac{1}{\theta_1}\left(f(y^{k+1}) - f(x^k)\right) - \frac{1}{\mathcal{L}_1\theta_1}\|g^k + h^k - \nabla f(x^k)\|^2,
\end{aligned}
$$

where the first inequality comes from $L_f$-smoothness of $f$, the second inequality comes from $L_f \leq \mathcal{L}_1$, and the last inequality comes from Young's inequality.

## C.3 Proof of Lemma B.3

First, from (9) and $\sigma_1 = \frac{\mu_f}{2\mathcal{L}_1}$, we have

$$
\begin{aligned}
g^k + h^k &= \frac{\mathcal{L}_1}{\eta}(\tilde{z}^k - \tilde{z}^{k+1}) + \mathcal{L}_1\sigma_1(\tilde{x}^k - \tilde{z}^{k+1}) - \partial\psi(z^{k+1}) \\
&= \frac{\mathcal{L}_1}{\eta}(\tilde{z}^k - \tilde{z}^{k+1}) + \frac{\mu_f}{2}(\tilde{x}^k - \tilde{z}^{k+1}) - \partial\psi(z^{k+1}),
\end{aligned}
$$

which implies that

$$
\begin{aligned}
\langle g^k + h^k, z^{k+1} - x^* \rangle &= \frac{\mu_f}{2}\langle z^{k+1} - x^*, \tilde{x}^k - \tilde{z}^{k+1}\rangle + \frac{\mathcal{L}_1}{\eta}\langle z^{k+1} - x^*, \tilde{z}^k - \tilde{z}^{k+1}\rangle \\
&\quad - \langle z^{k+1} - x^*, \partial\psi(z^{k+1})\rangle \\
&\leq \frac{\mu_f}{2}\langle z^{k+1} - x^*, \tilde{x}^k - \tilde{z}^{k+1}\rangle + \frac{\mathcal{L}_1}{\eta}\langle z^{k+1} - x^*, \tilde{z}^k - \tilde{z}^{k+1}\rangle \\
&\quad + \psi(x^*) - \psi(z^{k+1}) - \frac{\mu_\psi}{2}\|z^{k+1} - x^*\|^2 \\
&= \frac{\mu_f}{2}\langle \tilde{z}^{k+1} - x^*, \tilde{x}^k - \tilde{z}^{k+1}\rangle + \frac{\mathcal{L}_1}{\eta}\langle \tilde{z}^{k+1} - x^*, \tilde{z}^k - \tilde{z}^{k+1}\rangle \\
&\quad + \psi(x^*) - \psi(z^{k+1}) - \frac{\mu_\psi}{2}\|z^{k+1} - x^*\|^2 \\
&\quad + \frac{\mu_f}{2}\langle z^{k+1} - \tilde{z}^{k+1}, \tilde{x}^k - \tilde{z}^{k+1}\rangle + \frac{\mathcal{L}_1}{\eta}\langle z^{k+1} - \tilde{z}^{k+1}, \tilde{z}^k - \tilde{z}^{k+1}\rangle \\
&= \frac{\mu_f}{4}\left(\|\tilde{x}^k - x^*\|^2 - \|\tilde{z}^{k+1} - x^*\|^2 - \|\tilde{x}^k - \tilde{z}^{k+1}\|^2\right) \\
&\quad + \frac{\mathcal{L}_1}{2\eta}\left(\|\tilde{z}^k - x^*\|^2 - \|\tilde{z}^{k+1} - x^*\|^2 - \|\tilde{z}^k - \tilde{z}^{k+1}\|^2\right) \\
&\quad + \frac{\mu_f}{4}\left(\|z^{k+1} - \tilde{z}^{k+1}\|^2 + \|\tilde{x}^k - \tilde{z}^{k+1}\|^2 - \|\tilde{x}^k - z^{k+1}\|^2\right) \\
&\quad + \frac{\mathcal{L}_1}{2\eta}\left(\|z^{k+1} - \tilde{z}^{k+1}\|^2 + \|\tilde{z}^k - \tilde{z}^{k+1}\|^2 - \|\tilde{z}^k - z^{k+1}\|^2\right) \\
&\quad + \psi(x^*) - \psi(z^{k+1}) - \frac{\mu_\psi}{2}\|z^{k+1} - x^*\|^2 \\
&\leq -\left(\frac{\mathcal{L}_1}{2\eta} + \frac{\mu_f}{4}\right)\|\tilde{z}^{k+1} - x^*\|^2 + \frac{\mathcal{L}_1}{2\eta}\|\tilde{z}^k - x^*\|^2 + \frac{\mu_f}{4}\|\tilde{x}^k - x^*\|^2 \\
&\quad + \left(\frac{\mathcal{L}_1}{2\eta} + \frac{\mu_f}{4}\right)\|z^{k+1} - \tilde{z}^{k+1}\|^2 - \frac{\mathcal{L}_1}{2\eta}\|\tilde{z}^k - z^{k+1}\|^2 \\
&\quad + \psi(x^*) - \psi(z^{k+1}) - \frac{\mu_\psi}{2}\|z^{k+1} - x^*\|^2,
\end{aligned}
$$

where in the first inequality we use that $\psi$ is $\mu_\psi$-strongly convex and in the last inequality we use $-\|\tilde{x}^k - z^{k+1}\|^2 \leq 0$.

For $\|\tilde{x}^k - x^*\|^2$, $\|\tilde{z}^k - z^{k+1}\|^2$, and $\|z^{k+1} - x^*\|^2$, from Young's inequality, we have

$$
\|\tilde{x}^k - x^*\|^2 \leq 2\|\tilde{x}^k - x^k\|^2 + 2\|x^k - x^*\|^2, \qquad \|\tilde{z}^k - z^{k+1}\|^2 \geq \frac{1}{2}\|z^k - z^{k+1}\|^2 - \|z^k - \tilde{z}^k\|^2,
$$

and

$$
\|z^{k+1} - x^*\|^2 \geq \frac{1}{2}\|\tilde{z}^{k+1} - x^*\|^2 - \|z^{k+1} - \tilde{z}^{k+1}\|^2.
$$

Hence, we arrive at

$$
\begin{aligned}
\langle g^k + h^k, z^{k+1} - x^* \rangle \quad \leq \quad & -\left( \frac{\mathcal{L}_1}{2\eta} + \frac{\mu_f}{4} + \frac{\mu_\psi}{4} \right) \|\tilde{z}^{k+1} - x^*\|^2 + \frac{\mathcal{L}_1}{2\eta} \|\tilde{z}^k - x^*\|^2 + \frac{\mu_f}{2} \|x^k - x^*\|^2 \\
& + \frac{\mu_f}{2} \|\tilde{x}^k - x^k\|^2 + \left( \frac{\mathcal{L}_1}{2\eta} + \frac{\mu_f}{4} + \frac{\mu_\psi}{2} \right) \|z^{k+1} - \tilde{z}^{k+1}\|^2 \\
& - \frac{\mathcal{L}_1}{4\eta} \|z^k - z^{k+1}\|^2 + \frac{\mathcal{L}_1}{2\eta} \|z^k - \tilde{z}^k\|^2 + \psi(x^*) - \psi(z^{k+1}) \\
= \quad & -\left( \frac{\mathcal{L}_1}{2\eta} + \frac{\mu}{4} \right) \|\tilde{z}^{k+1} - x^*\|^2 + \frac{\mathcal{L}_1}{2\eta} \|\tilde{z}^k - x^*\|^2 + \frac{\mu_f}{2} \|x^k - x^*\|^2 \\
& + \left( \frac{\mathcal{L}_1}{2\eta} + \frac{\mu_f}{2} \right) \|e^k\|^2 + \left( \frac{\mathcal{L}_1}{2\eta} + \frac{\mu}{2} \right) \|e^{k+1}\|^2 \\
& - \frac{\mathcal{L}_1}{4\eta} \|z^k - z^{k+1}\|^2 + \psi(x^*) - \psi(z^{k+1}) \\
= \quad & -\tilde{\mathcal{Z}}^{k+1} + \frac{\mathcal{L}_1 \tilde{\mathcal{Z}}^k}{\mathcal{L}_1 + \eta\mu/2} + \frac{\mu_f}{2} \|x^k - x^*\|^2 + \left( \frac{\mathcal{L}_1}{2\eta} + \frac{\mu_f}{2} \right) \|e^k\|^2 \\
& + \left( \frac{\mathcal{L}_1}{2\eta} + \frac{\mu}{2} \right) \|e^{k+1}\|^2 - \frac{\mathcal{L}_1}{4\eta} \|z^k - z^{k+1}\|^2 + \psi(x^*) - \psi(z^{k+1}).
\end{aligned}
$$

## C.4  Proof of Lemma 3.3

Since $\theta_1 + \theta_2 \leq 1$, and $f$ is $\mu_f$-strong convex, we have

$$
\begin{aligned}
f(x^*) \quad \geq \quad & f(x^k) + \langle \nabla f(x^k), x^* - x^k \rangle + \frac{\mu_f}{2} \|x^k - x^*\|^2 \\
= \quad & f(x^k) + \frac{\mu_f}{2} \|x^k - x^*\|^2 + \langle \nabla f(x^k), x^* - z^k + z^k - x^k \rangle \\
= \quad & f(x^k) + \frac{\mu_f}{2} \|x^k - x^*\|^2 + \langle \nabla f(x^k), x^* - z^k \rangle + \frac{\theta_2}{\theta_1} \langle \nabla f(x^k), x^k - w^k \rangle + \frac{1 - \theta_1 - \theta_2}{\theta_1} \langle \nabla f(x^k), x^k - y^k \rangle \\
= \quad & f(x^k) + \frac{\theta_2}{\theta_1} \langle \nabla f(x^k), x^k - w^k \rangle + \frac{1 - \theta_1 - \theta_2}{\theta_1} (f(y^k) - f(x^k) - \langle \nabla f(x^k), y^k - x^k \rangle) \\
& + \frac{1 - \theta_1 - \theta_2}{\theta_1} (f(x^k) - f(y^k)) + \mathbb{E}_k \left[ \frac{\mu_f}{2} \|x^k - x^*\|^2 + \langle g^k + h^k, x^* - z^{k+1} \rangle + \langle g^k + h^k, z^{k+1} - z^k \rangle \right],
\end{aligned}
$$

where the last equality follows from the convexity of $f$ and $\mathbb{E}_k[g^k + h^k] = \nabla f(x^k)$. For the last term in the above equality, we have

$$
\begin{aligned}
& \mathbb{E}_k \left[ \frac{\mu_f}{2} \|x^k - x^*\|^2 + \langle g^k + h^k, x^* - z^{k+1} \rangle + \langle g^k + h^k, z^{k+1} - z^k \rangle - \psi(z^{k+1}) + \psi(x^*) - \tilde{\mathcal{Z}}^{k+1} \right] \\
\overset{(18)}{\geq} \quad & -\frac{\mathcal{L}_1 \tilde{\mathcal{Z}}^k}{\mathcal{L}_1 + \eta\mu/2} + \mathbb{E}_k \left[ \langle g^k + h^k, z^{k+1} - z^k \rangle + \frac{\mathcal{L}_1}{4\eta} \|z^k - z^{k+1}\|^2 \right] \\
& - \left( \frac{\mathcal{L}_1}{2\eta} + \frac{\mu_f}{2} \right) \|e^k\|^2 - \left( \frac{\mathcal{L}_1}{2\eta} + \frac{\mu}{2} \right) \mathbb{E}_k \|e^{k+1}\|^2 \\
\overset{(17)}{\geq} \quad & -\frac{\mathcal{L}_1 \tilde{\mathcal{Z}}^k}{\mathcal{L}_1 + \eta\mu/2} - \left( \frac{\mathcal{L}_1}{2\eta} + \frac{\mu_f}{2} \right) \|e^k\|^2 - \left( \frac{\mathcal{L}_1}{2\eta} + \frac{\mu}{2} \right) \mathbb{E}_k \|e^{k+1}\|^2 \\
& + \mathbb{E}_k \left[ \frac{1}{\theta_1} (f(y^{k+1}) - f(x^k)) - \frac{1}{\mathcal{L}_1 \theta_1} \|g^k + h^k - \nabla f(x^k)\|^2 \right] \\
\overset{(16)}{\geq} \quad & -\frac{\mathcal{L}_1 \tilde{\mathcal{Z}}^k}{\mathcal{L}_1 + \eta\mu/2} - \left( \frac{\mathcal{L}_1}{2\eta} + \frac{\mu_f}{2} \right) \|e^k\|^2 - \left( \frac{\mathcal{L}_1}{2\eta} + \frac{\mu}{2} \right) \mathbb{E}_k \|e^{k+1}\|^2 \\
& + \mathbb{E}_k \left[ \frac{1}{\theta_1} (f(y^{k+1}) - f(x^k)) - \frac{2L}{n\mathcal{L}_1 \theta_1} (f(w^k) - f(x^k) - \langle \nabla f(x^k), w^k - x^k \rangle) \right].
\end{aligned}
$$

Therefore,

$$
\mathbb{E}_k \left[ f(x^*) - \psi(z^{k+1}) + \psi(x^*) - \tilde{\mathcal{Z}}^{k+1} \right] + \left( \frac{\mathcal{L}_1}{2\eta} + \frac{\mu_f}{2} \right) \|e^k\|^2 + \left( \frac{\mathcal{L}_1}{2\eta} + \frac{\mu}{2} \right) \mathbb{E}_k \|e^{k+1}\|^2
$$

$$
- \frac{1 - \theta_1 - \theta_2}{\theta_1} (f(y^k) - f(x^k) - \langle \nabla f(x^k), y^k - x^k \rangle)
$$

$$
\geq \quad - \frac{\mathcal{L}_1 \tilde{\mathcal{Z}}^k}{\mathcal{L}_1 + \eta\mu/2} - \frac{1 - \theta_1 - \theta_2}{\theta_1} f(y^k) + \frac{1}{\theta_1} \mathbb{E}_k[f(y^{k+1})] - \frac{\theta_2}{\theta_1} \left( f(x^k) + \langle \nabla f(x^k), w^k - x^k \rangle \right)
$$

$$
- \frac{2L}{n\mathcal{L}_1 \theta_1} (f(w^k) - f(x^k) - \langle \nabla f(x^k), w^k - x^k \rangle)
$$

$$
= \quad - \frac{\mathcal{L}_1 \tilde{\mathcal{Z}}^k}{\mathcal{L}_1 + \eta\mu/2} - \frac{1 - \theta_1 - \theta_2}{\theta_1} f(y^k) + \frac{1}{\theta_1} \mathbb{E}_k[f(y^{k+1})] - \frac{\theta_2}{\theta_1} f(w^k)
$$

$$
+ \frac{1}{\theta_1} \left( \theta_2 - \frac{2L}{n\mathcal{L}_1} \right) (f(w^k) - f(x^k) - \langle \nabla f(x^k), w^k - x^k \rangle).
$$

From the convexity of $\psi$, and

$$
y^{k+1} = x^k + \theta_1 (z^{k+1} - z^k) = \theta_1 z^{k+1} + \theta_2 w^k + (1 - \theta_1 - \theta_2) y^k,
$$

we have

$$
\psi(z^{k+1}) \geq \frac{1}{\theta_1} \psi(y^{k+1}) - \frac{\theta_2}{\theta_1} \psi(w^k) - \frac{1 - \theta_1 - \theta_2}{\theta_1} \psi(y^k).
$$

Hence, we can obtain

$$
P(x^*) + \left( \frac{\mathcal{L}_1}{2\eta} + \frac{\mu_f}{2} \right) \|e^k\|^2 + \left( \frac{\mathcal{L}_1}{2\eta} + \frac{\mu}{2} \right) \mathbb{E}_k \|e^{k+1}\|^2 - \frac{1 - \theta_1 - \theta_2}{\theta_1} (f(y^k) - f(x^k) - \langle \nabla f(x^k), y^k - x^k \rangle)
$$

$$
\geq \quad \mathbb{E}_k[\tilde{\mathcal{Z}}^{k+1}] - \frac{\mathcal{L}_1 \tilde{\mathcal{Z}}^k}{\mathcal{L}_1 + \eta\mu/2} - \frac{1 - \theta_1 - \theta_2}{\theta_1} P(y^k) + \frac{1}{\theta_1} \mathbb{E}_k[P(y^{k+1})] - \frac{\theta_2}{\theta_1} P(w^k)
$$

$$
+ \frac{1}{\theta_1} \left( \theta_2 - \frac{2L}{n\mathcal{L}_1} \right) (f(w^k) - f(x^k) - \langle \nabla f(x^k), w^k - x^k \rangle).
$$

After rearranging we can get the result.

# D  Proofs of Lemma 3.4, Lemma 3.5, Lemma B.4, and Lemma B.5

## D.1  Proof of Lemma 3.4

First, we have

$$
\begin{aligned}
&\mathbb{E}_k[\|e_\tau^{k+1}\|^2] \\
&\overset{(1)}{\leq} (1-\delta)\mathbb{E}_k\|e_\tau^k + \frac{\eta}{\mathcal{L}_1}g_\tau^k\|^2 \\
&= (1-\delta)\mathbb{E}_k\|e_\tau^k + \frac{\eta}{\mathcal{L}_1}(\nabla f^{(\tau)}(x^k) - \nabla f^{(\tau)}(w^k)) + \frac{\eta}{\mathcal{L}_1}g_\tau^k - \frac{\eta}{\mathcal{L}_1}(\nabla f^{(\tau)}(x^k) - \nabla f^{(\tau)}(w^k))\|^2 \\
&= (1-\delta)\mathbb{E}_k\|e_\tau^k + \frac{\eta}{\mathcal{L}_1}(\nabla f^{(\tau)}(x^k) - \nabla f^{(\tau)}(w^k) + \nabla f^{(\tau)}(w^k) - h_\tau^k)\|^2 \\
&\quad + \frac{(1-\delta)\eta^2}{\mathcal{L}_1^2}\mathbb{E}_k\|\nabla f_{i_k^\tau}^{(\tau)}(x^k) - \nabla f_{i_k^\tau}^{(\tau)}(w^k) - (\nabla f^{(\tau)}(x^k) - \nabla f^{(\tau)}(w^k))\|^2 \\
&\leq (1-\delta)\|e_\tau^k + \frac{\eta}{\mathcal{L}_1}(\nabla f^{(\tau)}(x^k) - h_\tau^k)\|^2 + \frac{(1-\delta)\eta^2}{\mathcal{L}_1^2}\mathbb{E}_k\|\nabla f_{i_k^\tau}^{(\tau)}(x^k) - \nabla f_{i_k^\tau}^{(\tau)}(w^k)\|^2 \\
&\leq (1-\delta)(1+\beta)\|e_\tau^k\|^2 + (1-\delta)\left(1 + \frac{1}{\beta}\right)\frac{\eta^2}{\mathcal{L}_1^2}\|\nabla f^{(\tau)}(x^k) - h_\tau^k\|^2 \\
&\quad + \frac{(1-\delta)\eta^2}{\mathcal{L}_1^2}\mathbb{E}_k\|\nabla f_{i_k^\tau}^{(\tau)}(x^k) - \nabla f_{i_k^\tau}^{(\tau)}(w^k)\|^2 \\
&\leq \left(1 - \frac{\delta}{2}\right)\|e_\tau^k\|^2 + \frac{2(1-\delta)\eta^2}{\delta\mathcal{L}_1^2}\|\nabla f^{(\tau)}(x^k) - h_\tau^k\|^2 + \frac{(1-\delta)\eta^2}{\mathcal{L}_1^2}\mathbb{E}_k\|\nabla f_{i_k^\tau}^{(\tau)}(x^k) - \nabla f_{i_k^\tau}^{(\tau)}(w^k)\|^2,
\end{aligned}
$$

where we use Young's inequality in the third inequality and choose $\beta = \frac{\delta}{2(1-\delta)}$ when $\delta < 1$. When $\delta = 1$, it is easy to see that the above inequality also holds.

Then from Young's inequality, we can obtain

$$
\begin{aligned}
&\frac{1}{n}\sum_{\tau=1}^{n}\mathbb{E}_k[\|e_\tau^{k+1}\|^2] \\
&\leq \left(1 - \frac{\delta}{2}\right)\frac{1}{n}\sum_{\tau=1}^{n}\|e_\tau^k\|^2 + \frac{4(1-\delta)\eta^2}{\delta n\mathcal{L}_1^2}\sum_{\tau=1}^{n}\|\nabla f^{(\tau)}(x^k) - \nabla f^{(\tau)}(w^k)\|^2 \\
&\quad + \frac{4(1-\delta)\eta^2}{\delta n\mathcal{L}_1^2}\sum_{\tau=1}^{n}\|\nabla f^{(\tau)}(w^k) - h_\tau^k\|^2 + \frac{(1-\delta)\eta^2}{n\mathcal{L}_1^2}\sum_{\tau=1}^{n}\mathbb{E}_k\|\nabla f_{i_k^\tau}^{(\tau)}(x^k) - \nabla f_{i_k^\tau}^{(\tau)}(w^k)\|^2 \\
&\overset{(14)}{\leq} \left(1 - \frac{\delta}{2}\right)\frac{1}{n}\sum_{\tau=1}^{n}\|e_\tau^k\|^2 + \frac{8(1-\delta)\eta^2\bar{L}}{\delta\mathcal{L}_1^2}\left(f(w^k) - f(x^k) - \langle\nabla f(x^k), w^k - x^k\rangle\right) \\
&\quad + \frac{4(1-\delta)\eta^2}{\delta n\mathcal{L}_1^2}\sum_{\tau=1}^{n}\|\nabla f^{(\tau)}(w^k) - h_\tau^k\|^2 + \frac{(1-\delta)\eta^2}{n\mathcal{L}_1^2}\sum_{\tau=1}^{n}\mathbb{E}_k\|\nabla f_{i_k^\tau}^{(\tau)}(x^k) - \nabla f_{i_k^\tau}^{(\tau)}(w^k)\|^2 \\
&\overset{(13)}{\leq} \left(1 - \frac{\delta}{2}\right)\frac{1}{n}\sum_{\tau=1}^{n}\|e_\tau^k\|^2 + \frac{4(1-\delta)\eta^2}{\delta n\mathcal{L}_1^2}\sum_{\tau=1}^{n}\|\nabla f^{(\tau)}(w^k) - h_\tau^k\|^2 \\
&\quad + \frac{2(1-\delta)\eta^2}{\mathcal{L}_1^2}\left(\frac{4\bar{L}}{\delta} + L\right)\left(f(w^k) - f(x^k) - \langle\nabla f(x^k), w^k - x^k\rangle\right).
\end{aligned}
$$

## D.2 Proof of Lemma 3.5

First, from the update rule of $w^k$, we have

$$\mathbb{E}_k[\|h_\tau^{k+1} - \nabla f^{(\tau)}(w^{k+1})\|^2]$$
$$= p\mathbb{E}_k[\|h_\tau^{k+1} - \nabla f^{(\tau)}(y^k)\|^2] + (1-p)\mathbb{E}_k[\|h_\tau^{k+1} - \nabla f^{(\tau)}(w^k)\|^2]$$
$$\leq p\left(1 + \frac{2p}{\delta_1}\right)\mathbb{E}_k[\|\nabla f^{(\tau)}(y^k) - \nabla f^{(\tau)}(w^k)\|^2] + p\left(1 + \frac{\delta_1}{2p}\right)\mathbb{E}_k[\|h_\tau^{k+1} - \nabla f^{(\tau)}(w^k)\|^2]$$
$$+ (1-p)\mathbb{E}_k[\|h_\tau^{k+1} - \nabla f^{(\tau)}(w^k)\|^2]$$
$$= p\left(1 + \frac{2p}{\delta_1}\right)\|\nabla f^{(\tau)}(y^k) - \nabla f^{(\tau)}(w^k)\|^2 + \left(1 + \frac{\delta_1}{2}\right)\mathbb{E}_k[\|h_\tau^{k+1} - \nabla f^{(\tau)}(w^k)\|^2]$$
$$\leq p\left(1 + \frac{2p}{\delta_1}\right)\|\nabla f^{(\tau)}(y^k) - \nabla f^{(\tau)}(w^k)\|^2 + \left(1 - \frac{\delta_1}{2}\right)\|h_\tau^k - \nabla f^{(\tau)}(w^k)\|^2,$$

where the first inequality comes from the Young's inequality and the last inequality comes from the contraction property of $Q_1$.

Then we can obtain

$$\frac{1}{n}\sum_{\tau=1}^n \mathbb{E}_k[\|h_\tau^{k+1} - \nabla f^{(\tau)}(w^{k+1})\|^2]$$
$$\leq \frac{p}{n}\left(1 + \frac{2p}{\delta_1}\right)\sum_{\tau=1}^n \|\nabla f^{(\tau)}(y^k) - \nabla f^{(\tau)}(w^k)\|^2 + \left(1 - \frac{\delta_1}{2}\right)\frac{1}{n}\sum_{\tau=1}^n \|h_\tau^k - \nabla f^{(\tau)}(w^k)\|^2$$
$$\leq \left(1 - \frac{\delta_1}{2}\right)\frac{1}{n}\sum_{\tau=1}^n \|h_\tau^k - \nabla f^{(\tau)}(w^k)\|^2 + \frac{2p}{n}\left(1 + \frac{2p}{\delta_1}\right)\sum_{\tau=1}^n \|\nabla f^{(\tau)}(y^k) - \nabla f^{(\tau)}(x^k)\|^2$$
$$+ \frac{2p}{n}\left(1 + \frac{2p}{\delta_1}\right)\sum_{\tau=1}^n \|\nabla f^{(\tau)}(w^k) - \nabla f^{(\tau)}(x^k)\|^2$$
$$\leq \left(1 - \frac{\delta_1}{2}\right)\frac{1}{n}\sum_{\tau=1}^n \|h_\tau^k - \nabla f^{(\tau)}(w^k)\|^2 + 4\bar{L}p\left(1 + \frac{2p}{\delta_1}\right)(f(y^k) - f(x^k) - \langle \nabla f(x^k), y^k - x^k \rangle)$$
$$+ 4\bar{L}p\left(1 + \frac{2p}{\delta_1}\right)(f(w^k) - f(x^k) - \langle \nabla f(x^k), w^k - x^k \rangle).$$

## D.3 Proof of Lemma B.4

Under Assumption 2.1, we have $\mathbb{E}[Q(x)] = \delta x$, and thus

$$\mathbb{E}_k\|e^{k+1}\|^2 = \mathbb{E}_k\left\|\frac{1}{n}\sum_{\tau=1}^n e_\tau^{k+1}\right\|^2$$
$$= \frac{1}{n^2}\sum_{i,j}\mathbb{E}_k\langle e_i^{k+1}, e_j^{k+1}\rangle$$
$$= \frac{1}{n^2}\sum_{\tau=1}^n \mathbb{E}_k\|e_\tau^{k+1}\|^2 + \frac{1}{n^2}\sum_{i\neq j}\mathbb{E}_k\langle e_i^{k+1}, e_j^{k+1}\rangle$$
$$\overset{(1)}{\leq} \frac{1-\delta}{n^2}\sum_{\tau=1}^n \mathbb{E}_k\left\|e_\tau^k + \frac{\eta}{\mathcal{L}_1}g_\tau^k\right\|^2 + \frac{(1-\delta)^2}{n^2}\sum_{i\neq j}\mathbb{E}_k\left\langle e_i^k + \frac{\eta}{\mathcal{L}_1}g_i^k, e_j^k + \frac{\eta}{\mathcal{L}_1}g_j^k\right\rangle$$
$$= \frac{(1-\delta)^2}{n^2}\mathbb{E}_k\left\|\sum_{\tau=1}^n (e_\tau^k + \frac{\eta}{\mathcal{L}_1}g_\tau^k)\right\|^2 + \frac{(1-\delta)\delta}{n^2}\sum_{\tau=1}^n \mathbb{E}_k\left\|e_\tau^k + \frac{\eta}{\mathcal{L}_1}g_\tau^k\right\|^2$$
$$\leq (1-\delta)\mathbb{E}_k\left\|e^k + \frac{\eta}{\mathcal{L}_1}g^k\right\|^2 + \frac{(1-\delta)\delta}{n^2}\sum_{\tau=1}^n \mathbb{E}_k\left\|e_\tau^k + \frac{\eta}{\mathcal{L}_1}g_\tau^k\right\|^2,$$

where we use the definitions of $e^k$ and $g^k$ in the last inequality. Then we can obtain

$$\mathbb{E}_k\|e^{k+1}\|^2$$

$$\leq \quad (1-\delta)\mathbb{E}_k\left\|e^k + \frac{\eta}{\mathcal{L}_1}g^k\right\|^2 + \frac{(1-\delta)\delta}{n^2}\sum_{\tau=1}^{n}\mathbb{E}_k\left\|e_\tau^k + \frac{\eta}{\mathcal{L}_1}g_\tau^k\right\|^2$$

$$\leq \quad (1-\delta)\mathbb{E}_k\left\|e^k + \frac{\eta}{\mathcal{L}_1}g^k\right\|^2 + \frac{2(1-\delta)\delta}{n^2}\sum_{\tau=1}^{n}\|e_\tau^k\|^2 + \frac{2(1-\delta)\delta\eta^2}{n^2\mathcal{L}_1^2}\sum_{\tau=1}^{n}\mathbb{E}_k\|g_\tau^k\|^2$$

$$\leq \quad (1-\delta)\mathbb{E}_k\left\|e^k + \frac{\eta}{\mathcal{L}_1}g^k\right\|^2 + \frac{2(1-\delta)\delta}{n^2}\sum_{\tau=1}^{n}\|e_\tau^k\|^2 + \frac{4(1-\delta)\delta\eta^2}{n^2\mathcal{L}_1^2}\sum_{\tau=1}^{n}\mathbb{E}_k\|\nabla f_{i_k^\tau}^{(\tau)}(x^k) - \nabla f_{i_k^\tau}^{(\tau)}(w^k)\|^2$$

$$+\frac{4(1-\delta)\delta\eta^2}{n^2\mathcal{L}_1^2}\sum_{\tau=1}^{n}\|\nabla f^{(\tau)}(w^k) - h_\tau^k\|^2$$

$$\overset{(13)}{\leq} \quad (1-\delta)\mathbb{E}_k\left\|e^k + \frac{\eta}{\mathcal{L}_1}g^k\right\|^2 + \frac{2(1-\delta)\delta}{n^2}\sum_{\tau=1}^{n}\|e_\tau^k\|^2 + \frac{4(1-\delta)\delta\eta^2}{n^2\mathcal{L}_1^2}\sum_{\tau=1}^{n}\|\nabla f^{(\tau)}(w^k) - h_\tau^k\|^2$$

$$+\frac{8(1-\delta)\delta L\eta^2}{n\mathcal{L}_1^2}\left(f(w^k) - f(x^k) - \langle\nabla f(x^k), w^k - x^k\rangle\right), \tag{20}$$

where in the second and third inequalities we use the Young's inequality.

For $(1-\delta)\mathbb{E}_k\left\|e^k + \frac{\eta}{\mathcal{L}_1}g^k\right\|^2$, we have

$$(1-\delta)\mathbb{E}_k\left\|e^k + \frac{\eta}{\mathcal{L}_1}g^k\right\|^2$$

$$= \quad (1-\delta)\mathbb{E}_k\left\|e^k + \frac{\eta}{\mathcal{L}_1}(\nabla f(x^k) - \nabla f(w^k)) + \frac{\eta}{\mathcal{L}_1}g^k - \frac{\eta}{\mathcal{L}_1}(\nabla f(x^k) - \nabla f(w^k))\right\|^2$$

$$= \quad (1-\delta)\mathbb{E}_k\left\|e^k + \frac{\eta}{\mathcal{L}_1}(\nabla f(x^k) - h^k)\right\|^2$$

$$+\frac{(1-\delta)\eta^2}{\mathcal{L}_1^2}\mathbb{E}_k\left\|\frac{1}{n}\sum_{\tau=1}^{n}\left(\nabla f_{i_k^\tau}^{(\tau)}(x^k) - \nabla f_{i_k^\tau}^{(\tau)}(w^k)\right) - (\nabla f(x^k) - \nabla f(w^k))\right\|^2$$

$$\leq \quad \left(1 - \frac{\delta}{2}\right)\|e^k\|^2 + \frac{2(1-\delta)\eta^2}{\delta\mathcal{L}_1^2}\|\nabla f(x^k) - h^k\|^2$$

$$+\frac{(1-\delta)\eta^2}{\mathcal{L}_1^2}\mathbb{E}_k\left\|\frac{1}{n}\sum_{\tau=1}^{n}\left(\nabla f_{i_k^\tau}^{(\tau)}(x^k) - \nabla f_{i_k^\tau}^{(\tau)}(w^k)\right) - (\nabla f(x^k) - \nabla f(w^k))\right\|^2$$

$$\overset{(16)}{\leq} \quad \left(1 - \frac{\delta}{2}\right)\|e^k\|^2 + \frac{2(1-\delta)\eta^2}{\delta\mathcal{L}_1^2}\|\nabla f(x^k) - h^k\|^2 + (1-\delta)\frac{2L\eta^2}{n\mathcal{L}_1^2}\left(f(w^k) - f(x^k) - \langle\nabla f(x^k), w^k - x^k\rangle\right).$$

Since $f$ is $L_f$-smooth, we have

$$\|\nabla f(x^k) - h^k\|^2 \quad \leq \quad 2\|\nabla f(x^k) - \nabla f(w^k)\|^2 + 2\|\nabla f(w^k) - h^k\|^2$$

$$\leq \quad 4L_f\left(f(w^k) - f(x^k) - \langle\nabla f(x^k), w^k - x^k\rangle\right) + 2\|\nabla f(w^k) - h^k\|^2.$$

Hence, we arrive at

$$(1-\delta)\mathbb{E}_k\left\|e^k + \frac{\eta}{\mathcal{L}_1}g^k\right\|^2 \quad \leq \quad \left(1 - \frac{\delta}{2}\right)\|e^k\|^2 + \frac{4(1-\delta)\eta^2}{\delta\mathcal{L}_1^2}\|\nabla f(w^k) - h^k\|^2$$

$$+\frac{2(1-\delta)\eta^2}{\mathcal{L}_1^2}\left(\frac{4L_f}{\delta} + \frac{L}{n}\right)\left(f(w^k) - f(x^k) - \langle\nabla f(x^k), w^k - x^k\rangle\right).$$

Combining (20) and the above inequality, we can obtain

$$
\mathbb{E}_k \|e^{k+1}\|^2
$$
$$
\leq \quad \left(1 - \frac{\delta}{2}\right) \|e^k\|^2 + \frac{2(1-\delta)\delta}{n^2} \sum_{\tau=1}^{n} \|e_\tau^k\|^2 + \frac{4(1-\delta)\delta\eta^2}{n^2 \mathcal{L}_1^2} \sum_{\tau=1}^{n} \|\nabla f^{(\tau)}(w^k) - h_\tau^k\|^2
$$
$$
+ \frac{2(1-\delta)\eta^2}{\mathcal{L}_1^2} \left(\frac{4L_f}{\delta} + \frac{L}{n} + \frac{4\delta L}{n}\right) \left(f(w^k) - f(x^k) - \langle \nabla f(x^k), w^k - x^k\rangle\right) + \frac{4(1-\delta)\eta^2}{\delta \mathcal{L}_1^2} \|\nabla f(w^k) - h^k\|^2
$$
$$
\leq \quad \left(1 - \frac{\delta}{2}\right) \|e^k\|^2 + \frac{2(1-\delta)\delta}{n^2} \sum_{\tau=1}^{n} \|e_\tau^k\|^2 + \frac{4(1-\delta)\delta\eta^2}{n^2 \mathcal{L}_1^2} \sum_{\tau=1}^{n} \|\nabla f^{(\tau)}(w^k) - h_\tau^k\|^2
$$
$$
+ \frac{2(1-\delta)\eta^2}{\mathcal{L}_1^2} \left(\frac{4L_f}{\delta} + \frac{5L}{n}\right) \left(f(w^k) - f(x^k) - \langle \nabla f(x^k), w^k - x^k\rangle\right) + \frac{4(1-\delta)\eta^2}{\delta \mathcal{L}_1^2} \|h^k - \nabla f(w^k)\|^2.
$$

### D.4    Proof of Lemma B.5

First, from the update rule of $w^k$, we can obtain

$$
\mathbb{E}_k \|h^{k+1} - \nabla f(w^{k+1})\|^2
$$
$$
= \quad p\mathbb{E}_k \|h^{k+1} - \nabla f(y^k)\|^2 + (1-p)\mathbb{E}_k \|h^{k+1} - \nabla f(w^k)\|^2
$$
$$
\leq \quad p\left(1 + \frac{2p}{\delta_1}\right) \|\nabla f(y^k) - \nabla f(w^k)\|^2 + p\left(1 + \frac{\delta_1}{2p}\right) \mathbb{E}_k \|h^{k+1} - \nabla f(w^k)\|^2 + (1-p)\mathbb{E}_k \|h^{k+1} - \nabla f(w^k)\|^2
$$
$$
= \quad p\left(1 + \frac{2p}{\delta_1}\right) \|\nabla f(y^k) - \nabla f(w^k)\|^2 + \left(1 + \frac{\delta_1}{2}\right) \mathbb{E}_k \|h^{k+1} - \nabla f(w^k)\|^2.
$$

Denote $q_\tau^k = h_\tau^{k+1} - \nabla f^{(\tau)}(w^k)$. For $\mathbb{E}_k \|h^{k+1} - \nabla f(w^k)\|^2$, under Assumption 2.1, we have $\mathbb{E}[Q_1(x)] = \delta_1 x$, and thus

$$
\mathbb{E}_k \|h^{k+1} - \nabla f(w^k)\|^2 \quad = \quad \mathbb{E}_k \left\| \frac{1}{n} \sum_{\tau=1}^{n} (h_\tau^{k+1} - \nabla f^{(\tau)}(w^k)) \right\|^2
$$
$$
= \quad \mathbb{E}_k \left\| \frac{1}{n} \sum_{\tau=1}^{n} q_\tau^k \right\|^2
$$
$$
= \quad \frac{1}{n^2} \sum_{i,j} \mathbb{E}_k \langle q_i^k, q_j^k \rangle
$$
$$
= \quad \frac{1}{n^2} \sum_{\tau=1}^{n} \mathbb{E}_k \|q_\tau^k\|^2 + \frac{1}{n^2} \sum_{i \neq j} \mathbb{E}_k \langle q_i^k, q_j^k \rangle
$$
$$
\leq \quad \frac{1 - \delta_1}{n^2} \sum_{\tau=1}^{n} \|h_\tau^k - \nabla f^{(\tau)}(w^k)\|^2 + \frac{(1-\delta_1)^2}{n^2} \sum_{i \neq j} \langle h_i^k - \nabla f^{(i)}(w^k), h_j^k - \nabla f^{(j)}(w^k) \rangle
$$
$$
= \quad \frac{(1-\delta_1)^2}{n^2} \left\| \sum_{\tau=1}^{n} (h_\tau^k - \nabla f^{(\tau)}(w^k)) \right\|^2 + \frac{(1-\delta_1)\delta_1}{n^2} \sum_{\tau=1}^{n} \|h_\tau^k - \nabla f^{(\tau)}(w^k)\|^2
$$
$$
\leq \quad (1-\delta_1) \|h^k - \nabla f(w^k)\|^2 + \frac{(1-\delta_1)\delta_1}{n^2} \sum_{\tau=1}^{n} \|h_\tau^k - \nabla f^{(\tau)}(w^k)\|^2,
$$

where in the first inequality we use the contraction property of $Q_1$ and the independence of compressors on different nodes, and in the last inequality we use $h^k = \frac{1}{n} \sum_{\tau=1}^{n} h_\tau^k$.

Hence, we arrive at

$$\mathbb{E}_k \| h^{k+1} - \nabla f(w^{k+1}) \|^2$$

$$\leq p \left( 1 + \frac{2p}{\delta_1} \right) \| \nabla f(y^k) - \nabla f(w^k) \|^2 + \left( 1 - \frac{\delta_1}{2} \right) \| h^k - \nabla f(w^k) \|^2$$

$$+ \frac{(1 - \frac{\delta_1}{2}) \delta_1}{n^2} \sum_{\tau=1}^{n} \| h_\tau^k - \nabla f^{(\tau)}(w^k) \|^2$$

$$\leq \left( 1 - \frac{\delta_1}{2} \right) \| h^k - \nabla f(w^k) \|^2 + \frac{(1 - \frac{\delta_1}{2}) \delta_1}{n^2} \sum_{\tau=1}^{n} \| h_\tau^k - \nabla f^{(\tau)}(w^k) \|^2$$

$$+ 2p \left( 1 + \frac{2p}{\delta_1} \right) \| \nabla f(y^k) - \nabla f(x^k) \|^2 + 2p \left( 1 + \frac{2p}{\delta_1} \right) \| \nabla f(w^k) - \nabla f(x^k) \|^2$$

$$\leq \left( 1 - \frac{\delta_1}{2} \right) \| h^k - \nabla f(w^k) \|^2 + \frac{(1 - \frac{\delta_1}{2}) \delta_1}{n^2} \sum_{\tau=1}^{n} \| h_\tau^k - \nabla f^{(\tau)}(w^k) \|^2$$

$$+ 4p L_f \left( 1 + \frac{2p}{\delta_1} \right) \left( f(y^k) - f(x^k) - \langle \nabla f(x^k), y^k - x^k \rangle \right)$$

$$+ 4p L_f \left( 1 + \frac{2p}{\delta_1} \right) \left( f(w^k) - f(x^k) - \langle \nabla f(x^k), w^k - x^k \rangle \right),$$

where we use Young's inequality in the second inequality.

# E  Proofs of Theorem 3.6, Theorem 3.7, and Theorem 3.8

## E.1  Proof of Theorem 3.6

From $\|e^k\|^2 \leq \frac{1}{n}\sum_{\tau=1}^{n}\|e_\tau^k\|^2$, Equation (10), and Lemma 3.3, we can obtain

$$
\begin{aligned}
&\mathbb{E}_k\left[\tilde{\mathcal{Z}}^{k+1} + \mathcal{Y}^{k+1} + \mathcal{W}^{k+1}\right] \\
\leq\ & \frac{\mathcal{L}_1 \tilde{\mathcal{Z}}^k}{\mathcal{L}_1 + \eta\mu/2} + (1 - \theta_1 - \theta_2 + \frac{\theta_2}{q})\mathcal{Y}^k + (1 - p + pq)\mathcal{W}^k + \left(\frac{\mathcal{L}_1}{2\eta} + \frac{\mu_f}{2}\right)\frac{1}{n}\sum_{\tau=1}^{n}\|e_\tau^k\|^2 \\
& + \left(\frac{\mathcal{L}_1}{2\eta} + \frac{\mu}{2}\right)\mathbb{E}_k\left[\frac{1}{n}\sum_{\tau=1}^{n}\|e_\tau^{k+1}\|^2\right] - \frac{1}{\theta_1}\left(\theta_2 - \frac{2L}{n\mathcal{L}_1}\right)(f(w^k) - f(x^k) - \langle\nabla f(x^k), w^k - x^k\rangle) \\
& - \frac{1 - \theta_1 - \theta_2}{\theta_1}(f(y^k) - f(x^k) - \langle\nabla f(x^k), y^k - x^k\rangle) \\
\overset{\text{Lemma 3.4}}{\leq}\ & \frac{\mathcal{L}_1 \tilde{\mathcal{Z}}^k}{\mathcal{L}_1 + \eta\mu/2} + (1 - \theta_1 - \theta_2 + \frac{\theta_2}{q})\mathcal{Y}^k + (1 - p + pq)\mathcal{W}^k + \left(\frac{\mathcal{L}_1}{\eta} + \mu\right)\frac{1}{n}\sum_{\tau=1}^{n}\|e_\tau^k\|^2 \\
& + \frac{4(1-\delta)\eta^2}{\delta n\mathcal{L}_1^2}\left(\frac{\mathcal{L}_1}{2\eta} + \frac{\mu}{2}\right)\sum_{\tau=1}^{n}\|\nabla f^{(\tau)}(w^k) - h_\tau^k\|^2 - \frac{1 - \theta_1 - \theta_2}{\theta_1}(f(y^k) - f(x^k) - \langle\nabla f(x^k), y^k - x^k\rangle) \\
& - \left(\frac{1}{\theta_1}\left(\theta_2 - \frac{2L}{n\mathcal{L}_1}\right) - \frac{2(1-\delta)\eta^2}{\mathcal{L}_1^2}\left(\frac{4\bar{L}}{\delta} + L\right)\left(\frac{\mathcal{L}_1}{2\eta} + \frac{\mu}{2}\right)\right)(f(w^k) - f(x^k) - \langle\nabla f(x^k), w^k - x^k\rangle) \\
\leq\ & \frac{\mathcal{L}_1 \tilde{\mathcal{Z}}^k}{\mathcal{L}_1 + \eta\mu/2} + (1 - \theta_1 - \theta_2 + \frac{\theta_2}{q})\mathcal{Y}^k + (1 - p + pq)\mathcal{W}^k + \frac{4\mathcal{L}_1}{3\eta}\frac{1}{n}\sum_{\tau=1}^{n}\|e_\tau^k\|^2 \\
& + \frac{8(1-\delta)}{9\theta_1\delta n\mathcal{L}_1}\sum_{\tau=1}^{n}\|\nabla f^{(\tau)}(w^k) - h_\tau^k\|^2 - \frac{1 - \theta_1 - \theta_2}{\theta_1}(f(y^k) - f(x^k) - \langle\nabla f(x^k), y^k - x^k\rangle) \\
& - \frac{1}{\theta_1}\left(\theta_2 - \frac{1}{\mathcal{L}_1}\left(\frac{2L}{n} + \frac{16(1-\delta)\bar{L}}{9\delta} + \frac{4(1-\delta)L}{9}\right)\right)(f(w^k) - f(x^k) - \langle\nabla f(x^k), w^k - x^k\rangle),
\end{aligned}
$$

where we use $\mu \leq \frac{\mathcal{L}_1}{3\eta}$ and $\eta = \frac{1}{3\theta_1}$ in the last inequality. Then, from Lemma 3.4, we have

$$
\begin{aligned}
&\mathbb{E}_k\left[\tilde{\mathcal{Z}}^{k+1} + \mathcal{Y}^{k+1} + \mathcal{W}^{k+1} + \frac{4\mathcal{L}_1}{\delta\eta}\cdot\frac{1}{n}\sum_{\tau=1}^{n}\|e^{k+1}\|^2\right] \\
\leq\ & \frac{\mathcal{L}_1 \tilde{\mathcal{Z}}^k}{\mathcal{L}_1 + \eta\mu/2} + (1 - \theta_1 - \theta_2 + \frac{\theta_2}{q})\mathcal{Y}^k + (1 - p + pq)\mathcal{W}^k + \left(1 - \frac{\delta}{6}\right)\frac{4\mathcal{L}_1}{\delta\eta}\cdot\frac{1}{n}\sum_{\tau=1}^{n}\|e^k\|^2 \\
& + \frac{56(1-\delta)}{9\theta_1\delta^2 n\mathcal{L}_1}\sum_{\tau=1}^{n}\|\nabla f^{(\tau)}(w^k) - h_\tau^k\|^2 - \frac{1 - \theta_1 - \theta_2}{\theta_1}(f(y^k) - f(x^k) - \langle\nabla f(x^k), y^k - x^k\rangle) \\
& - \frac{1}{\theta_1}\left(\theta_2 - \frac{1}{\mathcal{L}_1}\left(\frac{2L}{n} + \frac{112(1-\delta)\bar{L}}{9\delta^2} + \frac{28(1-\delta)L}{9\delta}\right)\right)(f(w^k) - f(x^k) - \langle\nabla f(x^k), w^k - x^k\rangle).
\end{aligned}
$$

Finally, by Lemma 3.5, we can get

$$
\mathbb{E}_k \left[ \tilde{\mathcal{Z}}^{k+1} + \mathcal{Y}^{k+1} + \mathcal{W}^{k+1} + \frac{4\mathcal{L}_1}{\delta\eta} \cdot \frac{1}{n} \sum_{\tau=1}^{n} \|e^{k+1}\|^2 + \frac{56(1-\delta)}{3\theta_1\delta^2\delta_1\mathcal{L}_1} \cdot \frac{1}{n} \sum_{\tau=1}^{n} \|h_\tau^{k+1} - \nabla f^{(\tau)}(w^{k+1})\|^2 \right]
$$

$$
\leq \quad \frac{\mathcal{L}_1\tilde{\mathcal{Z}}^k}{\mathcal{L}_1 + \eta\mu/2} + (1 - \theta_1 - \theta_2 + \frac{\theta_2}{q})\mathcal{Y}^k + (1 - p + pq)\mathcal{W}^k + \left(1 - \frac{\delta}{6}\right)\frac{4\mathcal{L}_1}{\delta\eta} \cdot \frac{1}{n} \sum_{\tau=1}^{n} \|e^k\|^2
$$

$$
+ \left(1 - \frac{\delta_1}{6}\right) \frac{56(1-\delta)}{3\theta_1\delta^2\delta_1\mathcal{L}_1} \cdot \frac{1}{n} \sum_{\tau=1}^{n} \|h_\tau^k - \nabla f^{(\tau)}(w^k)\|^2
$$

$$
- \frac{1}{\theta_1} \left(1 - \theta_1 - \theta_2 - \frac{224(1-\delta)\bar{L}p}{3\delta^2\delta_1\mathcal{L}_1}\left(1 + \frac{2p}{\delta_1}\right)\right)(f(y^k) - f(x^k) - \langle\nabla f(x^k), y^k - x^k\rangle)
$$

$$
- \frac{1}{\theta_1} \left(\theta_2 - \frac{1}{\mathcal{L}_1}\left(\frac{2L}{n} + \frac{112(1-\delta)\bar{L}}{9\delta^2} + \frac{28(1-\delta)L}{9\delta} + \frac{224(1-\delta)\bar{L}p}{3\delta^2\delta_1}\left(1 + \frac{2p}{\delta_1}\right)\right)\right)
$$

$$
\cdot (f(w^k) - f(x^k) - \langle\nabla f(x^k), w^k - x^k\rangle)
$$

$$
\leq \quad \frac{\mathcal{L}_1\tilde{\mathcal{Z}}^k}{\mathcal{L}_1 + \eta\mu/2} + (1 - \theta_1 - \theta_2 + \frac{\theta_2}{q})\mathcal{Y}^k + (1 - p + pq)\mathcal{W}^k + \left(1 - \frac{\delta}{6}\right)\frac{4\mathcal{L}_1}{\delta\eta} \cdot \frac{1}{n} \sum_{\tau=1}^{n} \|e^k\|^2
$$

$$
+ \left(1 - \frac{\delta_1}{6}\right) \frac{56(1-\delta)}{3\theta_1\delta^2\delta_1\mathcal{L}_1} \cdot \frac{1}{n} \sum_{\tau=1}^{n} \|h_\tau^k - \nabla f^{(\tau)}(w^k)\|^2
$$

$$
- \frac{1}{\theta_1} \left(1 - \theta_1 - \theta_2 - \frac{\mathcal{L}_2}{3\mathcal{L}_1}\right)(f(y^k) - f(x^k) - \langle\nabla f(x^k), y^k - x^k\rangle)
$$

$$
- \frac{1}{\theta_1} \left(\theta_2 - \frac{\mathcal{L}_2}{3\mathcal{L}_1}\right)(f(w^k) - f(x^k) - \langle\nabla f(x^k), w^k - x^k\rangle),
$$

where we use the definition of $\mathcal{L}_2$ in the last inequality. When $\theta_2 \geq \frac{\mathcal{L}_2}{3\mathcal{L}_1}$ and $\theta_1 + 2\theta_2 \leq 1$, we can get the result.

## E.2   Proof of Theorem 3.7

From Lemma 3.3 and (10), we have

$$\mathbb{E}_k\left[\tilde{\mathcal{Z}}^{k+1} + \mathcal{Y}^{k+1} + \mathcal{W}^{k+1}\right]$$

$$\leq \quad \frac{\mathcal{L}_1\tilde{\mathcal{Z}}^k}{\mathcal{L}_1 + \eta\mu/2} + \left(1 - \theta_1 - \theta_2 + \frac{\theta_2}{q}\right)\mathcal{Y}^k + (1 - p + pq)\mathcal{W}^k + \left(\frac{\mathcal{L}_1}{2\eta} + \frac{\mu_f}{2}\right)\|e^k\|^2$$

$$+ \left(\frac{\mathcal{L}_1}{2\eta} + \frac{\mu}{2}\right)\mathbb{E}_k\|e^{k+1}\|^2 - \frac{1}{\theta_1}\left(\theta_2 - \frac{2L}{n\mathcal{L}_1}\right)(f(w^k) - f(x^k) - \langle\nabla f(x^k), w^k - x^k\rangle)$$

$$- \frac{1 - \theta_1 - \theta_2}{\theta_1}(f(y^k) - f(x^k) - \langle\nabla f(x^k), y^k - x^k\rangle)$$

$$\overset{\text{Lemma } B.4}{\leq} \quad \frac{\mathcal{L}_1\tilde{\mathcal{Z}}^k}{\mathcal{L}_1 + \eta\mu/2} + \left(1 - \theta_1 - \theta_2 + \frac{\theta_2}{q}\right)\mathcal{Y}^k + (1 - p + pq)\mathcal{W}^k + \left(\frac{\mathcal{L}_1}{\eta} + \mu\right)\|e^k\|^2$$

$$+ \left(\frac{\mathcal{L}_1}{2\eta} + \frac{\mu}{2}\right)\frac{2(1 - \delta)\delta}{n} \cdot \frac{1}{n}\sum_{\tau=1}^{n}\|e_\tau^k\|^2 + \frac{8(1 - \delta)\delta\eta}{3n\mathcal{L}_1} \cdot \frac{1}{n}\sum_{\tau=1}^{n}\|h_\tau^k - \nabla f^{(\tau)}(w^k)\|^2$$

$$+ \frac{8(1 - \delta)\eta}{3\delta\mathcal{L}_1}\|h^k - \nabla f(w^k)\|^2 - \frac{1 - \theta_1 - \theta_2}{\theta_1}(f(y^k) - f(x^k) - \langle\nabla f(x^k), y^k - x^k\rangle)$$

$$- \left(\frac{1}{\theta_1}\left(\theta_2 - \frac{2L}{n\mathcal{L}_1}\right) - \frac{4(1 - \delta)\eta}{3\mathcal{L}_1}\left(\frac{4L_f}{\delta} + \frac{5L}{n}\right)\right)(f(w^k) - f(x^k) - \langle\nabla f(x^k), w^k - x^k\rangle)$$

$$\leq \quad \frac{6\theta_1\mathcal{L}_1\tilde{\mathcal{Z}}^k}{6\theta_1\mathcal{L}_1 + \mu} + \left(1 - \theta_1 - \theta_2 + \frac{\theta_2}{q}\right)\mathcal{Y}^k + (1 - p + pq)\mathcal{W}^k + \frac{4\mathcal{L}_1}{3\eta}\|e^k\|^2$$

$$+ \frac{4\mathcal{L}_1(1 - \delta)\delta}{3\eta n} \cdot \frac{1}{n}\sum_{\tau=1}^{n}\|e_\tau^k\|^2 + \frac{8(1 - \delta)\delta\eta}{3n\mathcal{L}_1} \cdot \frac{1}{n}\sum_{\tau=1}^{n}\|h_\tau^k - \nabla f^{(\tau)}(w^k)\|^2$$

$$+ \frac{8(1 - \delta)\eta}{3\delta\mathcal{L}_1}\|h^k - \nabla f(w^k)\|^2 - \frac{1 - \theta_1 - \theta_2}{\theta_1}(f(y^k) - f(x^k) - \langle\nabla f(x^k), y^k - x^k\rangle)$$

$$- \frac{1}{\theta_1}\left(\theta_2 - \frac{2L}{n\mathcal{L}_1} - \frac{4(1 - \delta)}{9\mathcal{L}_1}\left(\frac{4L_f}{\delta} + \frac{5L}{n}\right)\right)(f(w^k) - f(x^k) - \langle\nabla f(x^k), w^k - x^k\rangle),$$

where we use $\mu \leq \frac{\mathcal{L}_1}{3\eta}$ and $\eta = \frac{1}{3\theta_1}$ in the last two inequalities. Then, from Lemma 3.4 and Lemma B.4, we can get

$$\mathbb{E}_k\left[\tilde{\mathcal{Z}}^{k+1} + \mathcal{Y}^{k+1} + \mathcal{W}^{k+1} + \frac{4\mathcal{L}_1}{\delta\eta}\|e^{k+1}\|^2 + \frac{28\mathcal{L}_1(1 - \delta)}{\delta\eta n} \cdot \frac{1}{n}\sum_{\tau=1}^{n}\|e_\tau^{k+1}\|^2\right]$$

$$\leq \quad \frac{6\theta_1\mathcal{L}_1\tilde{\mathcal{Z}}^k}{6\theta_1\mathcal{L}_1 + \mu} + \left(1 - \theta_1 - \theta_2 + \frac{\theta_2}{q}\right)\mathcal{Y}^k + (1 - p + pq)\mathcal{W}^k + \left(1 - \frac{\delta}{6}\right)\frac{4\mathcal{L}_1}{\delta\eta}\|e^k\|^2$$

$$+ \left(1 - \frac{\delta}{6}\right)\frac{28\mathcal{L}_1(1 - \delta)}{\delta\eta n} \cdot \frac{1}{n}\sum_{\tau=1}^{n}\|e_\tau^k\|^2 + \frac{112(1 - \delta)\eta}{\delta^2 n\mathcal{L}_1} \cdot \frac{1}{n}\sum_{\tau=1}^{n}\|h_\tau^k - \nabla f^{(\tau)}(w^k)\|^2$$

$$+ \frac{56(1 - \delta)\eta}{3\delta^2\mathcal{L}_1}\|h^k - \nabla f(w^k)\|^2 - \frac{1 - \theta_1 - \theta_2}{\theta_1}(f(y^k) - f(x^k) - \langle\nabla f(x^k), y^k - x^k\rangle)$$

$$- \frac{1}{\theta_1}\left(\theta_2 - \frac{2L}{n\mathcal{L}_1} - \frac{28(1 - \delta)}{9\delta\mathcal{L}_1}\left(\frac{4L_f}{\delta} + \frac{5L}{n}\right) - \frac{56(1 - \delta)^2}{3\delta\mathcal{L}_1}\left(\frac{4\bar{L}}{\delta n} + \frac{L}{n}\right)\right)$$

$$\cdot(f(w^k) - f(x^k) - \langle\nabla f(x^k), w^k - x^k\rangle).$$

We simplify the coefficient of the last term in the above inequality as follows.

$$\frac{28(1 - \delta)}{9\delta\mathcal{L}_1}\left(\frac{4L_f}{\delta} + \frac{5L}{n}\right) + \frac{56(1 - \delta)^2}{3\delta\mathcal{L}_1}\left(\frac{4\bar{L}}{\delta n} + \frac{L}{n}\right) \leq \frac{112(1 - \delta)L_f}{9\delta^2\mathcal{L}_1} + \frac{224(1 - \delta)\bar{L}}{3\delta^2 n\mathcal{L}_1} + \frac{308(1 - \delta)L}{9\delta n\mathcal{L}_1}.$$

Finally, from Lemma 3.5 and Lemma B.5, we can obtain

$$
\mathbb{E}_k\left[\tilde{\mathcal{Z}}^{k+1} + \mathcal{Y}^{k+1} + \mathcal{W}^{k+1} + \frac{4\mathcal{L}_1}{\delta\eta}\|e^{k+1}\|^2 + \frac{28\mathcal{L}_1(1-\delta)}{\delta\eta n} \cdot \frac{1}{n}\sum_{\tau=1}^{n}\|e_\tau^{k+1}\|^2\right.
$$

$$
\left. + \frac{56(1-\delta)\eta}{\delta^2\delta_1\mathcal{L}_1}\|h^{k+1} - \nabla f(w^{k+1})\|^2 + \frac{504(1-\delta)\eta}{\delta^2\delta_1 n\mathcal{L}_1} \cdot \frac{1}{n}\sum_{\tau=1}^{n}\|h_\tau^{k+1} - \nabla f^{(\tau)}(w^{k+1})\|^2\right]
$$

$$
\leq \quad \frac{6\theta_1\mathcal{L}_1\tilde{\mathcal{Z}}^k}{6\theta_1\mathcal{L}_1 + \mu} + (1 - \theta_1 - \theta_2 + \frac{\theta_2}{q})\mathcal{Y}^k + (1 - p + pq)\mathcal{W}^k + \left(1 - \frac{\delta}{6}\right)\frac{4\mathcal{L}_1}{\delta\eta}\|e^k\|^2
$$

$$
+ \left(1 - \frac{\delta}{6}\right)\frac{28\mathcal{L}_1(1-\delta)}{\delta\eta n} \cdot \frac{1}{n}\sum_{\tau=1}^{n}\|e_\tau^k\|^2 + \left(1 - \frac{\delta_1}{6}\right)\frac{504(1-\delta)\eta}{\delta^2\delta_1 n\mathcal{L}_1} \cdot \frac{1}{n}\sum_{\tau=1}^{n}\|h_\tau^k - \nabla f^{(\tau)}(w^k)\|^2
$$

$$
+ \left(1 - \frac{\delta_1}{6}\right)\frac{56(1-\delta)\eta}{\delta^2\delta_1\mathcal{L}_1}\|h^k - \nabla f(w^k)\|^2 - \frac{1}{\theta_1}\left(1 - \theta_1 - \theta_2 - \frac{224(1-\delta)p}{3\delta^2\delta_1\mathcal{L}_1}\left(L_f + \frac{9\bar{L}}{n}\right)\left(1 + \frac{2p}{\delta_1}\right)\right)
$$

$$
\cdot(f(y^k) - f(x^k) - \langle\nabla f(x^k), y^k - x^k\rangle)
$$

$$
- \frac{1}{\theta_1}\left(\theta_2 - \frac{2L}{n\mathcal{L}_1} - \frac{112(1-\delta)L_f}{9\delta^2\mathcal{L}_1} - \frac{224(1-\delta)\bar{L}}{3\delta^2 n\mathcal{L}_1} - \frac{308(1-\delta)L}{9\delta n\mathcal{L}_1} - \frac{224(1-\delta)p}{3\delta^2\delta_1\mathcal{L}_1}\left(L_f + \frac{9\bar{L}}{n}\right)\left(1 + \frac{2p}{\delta_1}\right)\right)
$$

$$
\cdot(f(w^k) - f(x^k) - \langle\nabla f(x^k), w^k - x^k\rangle)
$$

$$
\leq \quad \Psi^k - \frac{1}{\theta_1}\left(1 - \theta_1 - \theta_2 - \frac{\mathcal{L}_3}{3\mathcal{L}_1}\right)(f(y^k) - f(x^k) - \langle\nabla f(x^k), y^k - x^k\rangle)
$$

$$
- \frac{1}{\theta_1}\left(\theta_2 - \frac{\mathcal{L}_3}{3\mathcal{L}_1}\right)(f(w^k) - f(x^k) - \langle\nabla f(x^k), w^k - x^k\rangle),
$$

where we use the definition of $\mathcal{L}_3$ in the last inequality. When $\theta_2 \geq \frac{\mathcal{L}_3}{3\mathcal{L}_1}$ and $\theta_1 + 2\theta_2 \leq 1$ we can get the result.

### E.3   Proof of Theorem 3.8

(i) First, we have $\frac{1}{3} \geq \theta_2 \geq \frac{\mathcal{L}_2}{3\mathcal{L}_1}$. Form the definition of $\theta_1$, we know $\theta_1 \leq \frac{1}{3}$. Then $\theta_1 + 2\theta_2 \leq 1$. Thus the result in Theorem 3.6 holds. Next we consider two cases:

**Case 1.** Suppose $L_f \leq \frac{\mathcal{L}_4}{p}$. In this case, we have $\theta_1 = \min\left(\sqrt{\frac{\mu}{\mathcal{L}_4 p}}\theta_2, \theta_2\right)$ and $\theta_2 = \frac{\mathcal{L}_4}{3\max\{L_f, \mathcal{L}_4\}} \geq \frac{p}{3}$.

**Case 1.1.** Suppose $\frac{\mu}{\mathcal{L}_4 p} \geq 1$. In this subcase, $\theta_1 = \theta_2$.

For $\frac{\mu}{\mu + 6\theta_1\mathcal{L}_1}$, we discuss two cases. If $\mathcal{L}_1 = \max(\mathcal{L}_4, L_f, 3\mu\eta) = 3\mu\eta$, then $\frac{\mu}{\mu + 6\theta_1\mathcal{L}_1} = \frac{1}{7} \geq \frac{p}{7}$. If $\mathcal{L}_1 = \max(\mathcal{L}_4, L_f, 3\mu\eta) = \max(\mathcal{L}_4, L_f)$, then $\mathcal{L}_1 = \frac{\mathcal{L}_4}{3\theta_2} = \frac{\mathcal{L}_4}{3\theta_1}$, which implies that $\frac{\mu}{\mu + 6\theta_1\mathcal{L}_1} = \frac{\mu}{\mu + 2\mathcal{L}_4} \geq \frac{p}{p+2} \geq \frac{p}{3}$.

By choosing $q = \frac{2}{3}$, we have $\theta_1 + \theta_2 - \frac{\theta_2}{q} = \frac{\theta_2}{2} \geq \frac{p}{6}$ and $p(1 - q) = \frac{p}{3}$.

**Case 1.2.** Suppose $\frac{\mu}{\mathcal{L}_4 p} < 1$. In this subcase, $\theta_1 = \sqrt{\frac{\mu}{\mathcal{L}_4 p}}\theta_2$.

For $\frac{\mu}{\mu + 6\theta_1\mathcal{L}_1}$, we discuss two cases. If $\mathcal{L}_1 = \max(\mathcal{L}_4, L_f, 3\mu\eta) = 3\mu\eta$, then $\frac{\mu}{\mu + 6\theta_1\mathcal{L}_1} = \frac{1}{7} \geq \frac{p}{7}$. If $\mathcal{L}_1 = \max(\mathcal{L}_4, L_f, 3\mu\eta) = \max(\mathcal{L}_4, L_f)$, then $\mathcal{L}_1 = \frac{\mathcal{L}_4}{3\theta_2} = \frac{1}{3\theta_1}\sqrt{\frac{\mu\mathcal{L}_4}{p}}$, which yields $\frac{\mu}{\mu + 6\theta_1\mathcal{L}_1} = \frac{\sqrt{\mu p/\mathcal{L}_4}}{\sqrt{\mu p/\mathcal{L}_4} + 2}$. Noticing that $\frac{\mu p}{\mathcal{L}_4} = \frac{\mu}{\mathcal{L}_4 p} \cdot p^2 < p^2 \leq 1$, we have $\frac{\mu}{\mu + 6\theta_1\mathcal{L}_1} \geq \frac{1}{3}\sqrt{\frac{\mu p}{\mathcal{L}_4}}$.

By choosing $q = 1 - \frac{1}{3}\sqrt{\frac{\mu}{\mathcal{L}_4 p}} \geq \frac{2}{3}$ so that $3\theta_2(1 - q) = \theta_1$, we have $\theta_1 + \theta_2 - \frac{\theta_2}{q} = \theta_1(1 - \frac{1}{3q}) \geq \frac{\theta_1}{2} = \frac{\theta_2}{2}\sqrt{\frac{\mu}{\mathcal{L}_4 p}} \geq \frac{1}{6}\sqrt{\frac{\mu p}{\mathcal{L}_4}}$ and $p(1 - q) = \frac{1}{3}\sqrt{\frac{\mu p}{\mathcal{L}_4}}$.

**Case 2.** Suppose $L_f > \frac{\mathcal{L}_4}{p}$. In this case, $\theta_1 = \min\left(\sqrt{\frac{\mu}{L_f}}, \frac{p}{3}\right)$, $\max(L_f, \mathcal{L}_4) = L_f$, and $\theta_2 = \frac{\mathcal{L}_4}{3L_f} < \frac{p}{3}$.

**Case 2.1.** Suppose $\sqrt{\frac{\mu}{L_f}} \geq \frac{p}{3}$. In this subcase, $\theta_1 = \frac{p}{3}$.

For $\frac{\mu}{\mu+6\theta_1\mathcal{L}_1}$, we discuss two cases. If $\mathcal{L}_1 = \max(\mathcal{L}_4, L_f, 3\mu\eta) = 3\mu\eta$, then $\frac{\mu}{\mu+6\theta_1\mathcal{L}_1} = \frac{1}{7} \geq \frac{p}{7}$. If $\mathcal{L}_1 = \max(\mathcal{L}_4, L_f, 3\mu\eta) = \max(\mathcal{L}_4, L_f) = L_f$, then $\frac{\mu}{\mu+6\theta_1\mathcal{L}_1} = \frac{\mu}{\mu+2pL_f} \geq \frac{\mu}{\mu+18\mu/p} = \frac{p}{p+18} \geq \frac{p}{19}$.

Let $q = \frac{2}{3}$. Then $\theta_1 + \theta_2 - \frac{\theta_2}{q} = \theta_1 - \frac{\theta_2}{2} > \frac{p}{6}$ and $p(1-q) = \frac{p}{3}$.

**Case 2.2.** Suppose $\sqrt{\frac{\mu}{L_f}} < \frac{p}{3}$. In this subcase, $\theta_1 = \sqrt{\frac{\mu}{L_f}}$.

For $\frac{\mu}{\mu+6\theta_1\mathcal{L}_1}$, we discuss two cases. If $\mathcal{L}_1 = \max(\mathcal{L}_4, L_f, 3\mu\eta) = 3\mu\eta$, then $\frac{\mu}{\mu+6\theta_1\mathcal{L}_1} = \frac{1}{7} \geq \frac{p}{7}$. If $\mathcal{L}_1 = \max(\mathcal{L}_4, L_f, 3\mu\eta) = \max(\mathcal{L}_4, L_f) = L_f$, then $\frac{\mu}{\mu+6\theta_1\mathcal{L}_1} = \frac{\mu}{\mu+6\sqrt{\mu L_f}} = \frac{\sqrt{\mu/L_f}}{\sqrt{\mu/L_f}+6} \geq \frac{3}{19}\sqrt{\frac{\mu}{L_f}}$.

Let $q = 1 - \frac{1}{p}\sqrt{\frac{\mu}{L_f}} > \frac{2}{3}$ so that $1 - q = \frac{1}{p}\theta_1$. Then $\theta_1 + \theta_2 - \frac{\theta_2}{q} = \theta_1 - \frac{1}{pq}\theta_1\theta_2 > (1 - \frac{1}{3q})\theta_1 > \frac{\theta_1}{2} = \frac{1}{2}\sqrt{\frac{\mu}{L_f}}$ and $p(1-q) = \sqrt{\frac{\mu}{L_f}}$.

Therefore, we have $\mathbb{E}[\Phi^k] \leq \epsilon\Phi^0$ as long as

$$k \geq \mathcal{O}\left(\frac{1}{\delta} + \frac{1}{\delta_1} + \frac{1}{p} + \sqrt{\frac{L_f}{\mu}} + \sqrt{\frac{\mathcal{L}_4}{\mu p}}\right).$$

Since $\mathcal{L}_4 = \mathcal{L}_2$, we can get the results.

(ii) By using Theorem 3.7 and $\bar{L} \leq nL_f$, same as (i), we can get the results.