# OpenReview forum: "Error Compensated Distributed SGD Can Be Accelerated"
_NeurIPS.cc/2021/Conference — NeurIPS 2021 Poster_

### Official Review · Reviewer_iBtS · 2021-07-14

**Rating:** 5
**Confidence:** 2

**Summary:**

This paper considers communication efficient algorithms for SGD. In this model, there are $n$ total workers and each worker has a local smooth convex loss function $f_i$ on $R^d$. The objective is for a coordinator to find an approximate global minimum for an objective function that represents the average loss across all workers while minimizing communication between the workers and the coordinator.

A common approach is for each worker to compute a sketch on their local batch of gradients that allows the coordinator to recover of the top $k$ gradients in magnitude, ideally improving the rate of convergence. However, the challenge with this approach is that the estimator formed by the coordinator using the sketches sent from the workers may not be an unbiased estimator of the overall gradient, which allows error to accumulate over multiple iterations. This paper introduces provable accelerated gradient-methods, in the sense of Nesterov momentum for contractive compressors, which are a general class of stochastic functions that form estimators of the gradient, including the top $k$ estimator.

**Limitations And Societal Impact:**

I do not think there are any limitations and potential negative societal impact of their work that the authors need to address.

**Main Review:**

It seems the algorithm performs the usual uniform sampling technique to acquire a batch of gradients for each step. The algorithm then applies a contractive compressor in the standard way to improve communication efficiency over naive SGD. To address the bias incurred by the contractive compressor, the algorithm attempts to learn the added bias and subtract the bias incurred from the previous iterate before using a standard proximal operator to update the current iterate.

I believe existing approaches that achieve communication efficient SGD also use this framework to deal with error accumulation and feedback, e.g., FetchSGD [RPU+20] also achieves acceleration though it seems the convergence rate is better in this paper. What is the algorithmic novelty that allows the improved convergence rate? This should perhaps be clarified in the paper.

Though the theoretical contributions of the paper seem very relevant to the machine learning and stochastic optimization communities, I think the exposition of the paper could be greatly improved in general. For example, I think discussion or comparison to related works or techniques are sometimes overlooked [IRU+19, RPU+20] or too brief [KRSJ19]. I also think the current text description does not give enough intuition for the main algorithm.

[KRSJ19] Sai Praneeth Karimireddy, Quentin Rebjock, Sebastian U. Stich, Martin Jaggi: Error Feedback Fixes SignSGD and other Gradient Compression Schemes. ICML 2019: 3252-3261

[IRU+19] Nikita Ivkin, Daniel Rothchild, Enayat Ullah, Vladimir Braverman, Ion Stoica, Raman Arora:
Communication-efficient Distributed SGD with Sketching. NeurIPS 2019: 13144-13154

[RPU+20] Daniel Rothchild, Ashwinee Panda, Enayat Ullah, Nikita Ivkin, Ion Stoica, Vladimir Braverman, Joseph Gonzalez, Raman Arora: FetchSGD: Communication-Efficient Federated Learning with Sketching. ICML 2020: 8253-8265

**Time Spent Reviewing:**

10

---

> ### Author Response · Authors · 2021-08-09
> **Issue 1: Relation to FetchSGD**
>
> > Issue 1:
>
> I believe existing approaches that achieve communication efficient SGD also use this framework to deal with error accumulation and feedback, e.g., FetchSGD [RPU+20] also achieves acceleration though it seems the convergence rate is better in this paper. What is the algorithmic novelty that allows the improved convergence rate? This should perhaps be clarified in the paper.
>
> > Reply to Issue 1:
>
> While FetchSGD is *broadly* relevant to our work, it is not *closely* relevant. Let us outline the main differences.
>
> - Most importantly, the FetchSGD paper of Rothchild et al focuses on *nonconvex optimization*, while we focus on *strongly convex optimization* in our paper. These are two very different worlds that should not be compared. Algorithms are quite different, acceleration in these settings is obtained using very different techniques, and the rates are different. Moreover, the convergence rate of FetchSGD is weak compared to newer results that use even *non-accelerated* error compensation. Indeed, Theorem 1 in FetchSGD gives $O(1/T^{1/2})$ rate (or the worse rate of $O(1/T^{1/3})$ under an alternative but still very strong assumption), while standard error compensation techniques of Tang et al (2020) and Koloskova et al (2020) give $O(1/T^{2/3})$ rates. The fastest rate of non-accelerated error compensation for nonconvex optimization is obtained by the EF21 method of Richtarik et al (2021); it is $O(1/T)$, and this work does not use any run/iterate-dependent assumptions. In short, while FetchSGD is a method in a similar area as ours, it is only tangentially relevant to our work, and its rate was surpassed in recent works multiple times.
>
> - FetchSGD does *not* deal with arbitrary contractive compressors and instead works with *sketches*, which are randomized linear transformations. In particular, it works with a very particular sketch: the CountSketch. In contrast, we work with arbitrary contractive compressors, which offer much more flexibility. In fact, the literature on error compensation is characterized by the ability of this technique to compensate the error caused by any contractive compressor. This is also a very major difference as we explicitly aim to develop the fastest error compensation method in the strongly convex regime that works with arbitrary contractive compressors. So, FetchSGD is not our competitor; it is a complementary piece of work.
>
> - Moreover, CountSketch used in FetchSGD is *not* a contractive compressor in general (unless a second round of communication is performed, which is not what the authors do here). Instead, Rothchild et al make several strong assumptions on the iterates of the method to make their theory work: see their Assumption 1 on "heavy hitters", for example. This is a major weakness as such run-dependent assumptions naturally can't be checked a-priori, and may not even hold. In our work we do not make any such strong assumptions. As we mentioned above, the rates are very weak compared to most recent work.
>
> - Finally, the momentum used in CountSketch does not help to accelerate their rates in theory. This is apparent already from the fact that the non-accelerated error compensated methods offer better rates. In contrast, the type of momentum we use leads to better rates, in fact, to current SOTA rates among all error compensated methods in the strongly convex regime.
>
> In summary, FetchSGD focus on nonconvex optimization, use strong iterate-dependent assumptions, work for a particular type of sketch only (which is not a contractive compressor the way they use it), and obtain rates that are weak compared to latest results in this area. No theoretical acceleration is obtained from their momentum. In contrast, we focus on strongly convex optimization, which is a different field altogether, do not rely on any iterate dependent assumptions, work with general contractive compressors, and obtain new SOTA rates among all error compensation methods.
>
> Algorithmic novelty, in brief:
>
> We managed to find a way to combine loopless Katyusha style momentum with error compensation in a way that makes both tricks work in tandem, with provable improvements. Lopless Katyusha, which is an accelerated, stochastic, and variance-reduced method, and achieve the optimal complexity among randomized methods for solving finite-sum strongly convex optimization problems. It was important to use the loopless version of Katyusha, rather than original Katyusha (or Nesterov or Polyak momentum, which we could not make to work), which is a relatively recent development, since this variant offers a much simpler proof, and we need proof simplicity if we want to combine acceleration with error compensation. We apply the contractive compression and error feedback to the distributed loopless Katyusha, and in order to control the compression error, we also used a learning scheme similar to than in VR-DIANA. We will clarify this in the revised version.
>
> References:
>
> - Tang et al (2020): Hanlin Tang, Xiangru Lian, Chen Yu, Tong Zhang, and Ji Liu. DoubleSqueeze: Parallel stochastic gradient descent with double-pass error-compensated compression. In Proceedings of the 36th International Conference on Machine Learning (ICML), 2020.
> - Koloskova et al (2020): Anastasia Koloskova, Tao Lin, S. Stich, and Martin Jaggi. Decentralized deep learning with arbitrary communication compression. In International Conference on Learning Representations (ICLR), 2020.
> - Richtarik et al (2021): Peter Richtarik, Igor Sokolov, Ilyas Fatkhullin. EF21: A New, Simpler, Theoretically Better, and Practically Faster Error Feedback, arXiv:2106.05203, 2021.

---

> ### Author Response · Authors · 2021-08-09
> **Issue 2: exposition can be improved**
>
> > Issue 2:
>
> Though the theoretical contributions of the paper seem very relevant to the machine learning and stochastic optimization communities, I think the exposition of the paper could be greatly improved in general. For example, I think discussion or comparison to related works or techniques are sometimes overlooked [IRU+19, RPU+20] or too brief [KRSJ19]. I also think the current text description does not give enough intuition for the main algorithm.
>
> [KRSJ19] Sai Praneeth Karimireddy, Quentin Rebjock, Sebastian U. Stich, Martin Jaggi: Error Feedback Fixes SignSGD and other Gradient Compression Schemes. ICML 2019: 3252-3261
>
> [IRU+19] Nikita Ivkin, Daniel Rothchild, Enayat Ullah, Vladimir Braverman, Ion Stoica, Raman Arora: Communication-efficient Distributed SGD with Sketching. NeurIPS 2019: 13144-13154
>
> [RPU+20] Daniel Rothchild, Ashwinee Panda, Enayat Ullah, Nikita Ivkin, Ion Stoica, Vladimir Braverman, Joseph Gonzalez, Raman Arora: FetchSGD: Communication-Efficient Federated Learning with Sketching. ICML 2020: 8253-8265
>
> > Reply to Issue 2:
>
> The reviewer suggested that there are two problems with exposition in our work:
>
> A) *That we should have cited some papers, such as [IRU+19, RPU+20], and that we did not mention in enough detail [KRSJ19].*
>
> Please note that we have a bit more discussion on related works in Section F in the appendix. Did you see this section?
>
> We note that *no reason* was provided for *why* we should have cited [IRU+19, RPU+20]. We already explained in detail by addressing Issue 1 in a separate post that [RPU+20] is simply not a closely relevant piece of work in the context of the focus of our paper. Yes, it is broadly relevant as it deals with communication efficient distributed optimization. But so are *hundreds* of other papers; and obviously we can't cite them all. So, we absolutely need to focus on citing the most closely relevant works only, plus a few selected more broadly relevant papers to chart the broader area.
>
> The [IRU+19] paper is also not closely relevant. If the reviewer believes this paper *is* relevant, then we kindly request detailed reasoning. We will of course cite any closely relevant work, and will be happy to receive such suggestions!
>
> Please let us know *what* do you believe we should have said about [KRSJ19], and why. Simply stating that we did not say enough is not very helpful to us. If we missed to say something of importance about this piece of work, something that has to be said in the context of our paper, please let us know what detail this is.
>
> B) *That we did not give enough intuitions about the algorithm*
>
> We describe the algorithm in Section 3.1. ECLK can loosely be seen as the combination of the error feedback technique, the learning scheme in VR-DIANA, and loopless Katyusha. We will certainly add more explanation to make this and other details clearer in the revised version.

---

> ### Author Response · Authors · 2021-08-09
> **Thanks**
>
> Dear reviewer iBtS,
>
> Thanks for your review and the effort you put into it.
>
>
> - We argue that the criticism that was raised is actually not valid, or is very very minor, as we explain in our response.
> - Please read our generic comment entitled "Thanks to all reviewers"
> - Please read our response to all reviewers. We believe you will see that no serious issues were raised by any reviewer - we managed to address them all satisfactorily.
>
> Since the issues you raised were all addressed as non-issues, or as very minor issues, we do not believe a score of 5 is appropriate. We believe our work is of a breakthrough nature and deserved marks in the 8-10 region.
>
> Thanks!
>
> Authors

---

> > ### Author Response · Authors · 2021-08-15
> > **Are you satisfied with our response to the 2 issues you raised?**
> >
> > Dear Reviewer iBtS,
> >
> > Are you satisfied with our response to the 2 issues you raised?
> >
> > - If yes, please would you consider raising your score appropriately?
> > - If not, please let us known which issues were not clarified sufficiently and why, so that we have a chance to respond.
> >
> > Best regards,
> >
> > Authors

---

> > ### Author Response · Authors · 2021-08-19
> > **Reminder: Are you satisfied with our response to the 2 issues you raised?**
> >
> > Dear Reviewer iBtS,
> >
> > Are you satisfied with our response to the 2 issues you raised?
> >
> > - If yes, please would you consider raising your score appropriately?
> > - If not, please let us known which issues were not clarified sufficiently and why, so that we have a chance to respond.
> >
> > Best regards,
> >
> > Authors

---

> > ### Author Response · Authors · 2021-08-23
> > **Second Reminder: Are you satisfied with our response to the 2 issues you raised?**
> >
> > Dear Reviewer iBtS,
> >
> > Are you satisfied with our response to the 2 issues you raised?
> >
> > - If yes, please would you consider raising your score appropriately?
> > - If not, please let us known which issues were not clarified sufficiently and why, so that we have a chance to respond.
> >
> > Best regards,
> >
> > Authors

---

> > ### Author Response · Authors · 2021-09-05
> > **Third reminder: Are you satisfied with our response to the 2 issues you raised?**
> >
> > Dear Reviewer iBtS,
> >
> > Are you satisfied with our response to the 2 issues you raised?
> >
> > - If yes, please would you consider raising your score appropriately?
> > - If not, please let us known which issues were not clarified sufficiently and why, so that we have a chance to respond.
> >
> > Best regards,
> >
> > Authors

---

### Official Review · Reviewer_ZXqE · 2021-07-16

**Rating:** 6
**Confidence:** 2

**Summary:**

The paper introduces an error compensation variant of the loopless Katyusha, ECLK. Theoretical convergence analysis is given and accelerated linear convergence rate is obtained using contraction operators. Experimental study of ECLK is given on the logistic regression problem for binary classification.

**Limitations And Societal Impact:**

The paper addressed the limitations and potential negative societal impact.

**Main Review:**

The paper applys error compensation to accelerated gradient methods, and introduces error compensated loopless Katyusha method. Convergence analysis is given of ECLK with contraction operators, and compared to ADIANA with unbiased compressor. Experimental results are given for binary classification with ECLK with three contraction compressors, and compared with other methods in literature.

The paper is clearly written and well organized.

**Update**: Thanks for the authors' response to the reviews. It helps with the clarification after reading the responses to all the reviewers.


**Time Spent Reviewing:**

3

---

> ### Author Response · Authors · 2021-08-08
> **Thanks!**
>
> Dear reviewer ZXqE,
>
> Thanks for recommending our paper to be accepted, albeit as a "borderline accept."
>
> - Please read our comment entitled "Thanks to all reviewers"
> - Please read our response to all reviewers. We believe you will see that no serious issues were raised by any reviewer - we managed to address them all satisfactorily.
> - Since you do not raise any issues, and because of the above, we would hope you could reconsider and give our work a higher score.
>
> Thanks!
>
> Authors

---

> > ### Author Response · Authors · 2021-08-15
> > **Did we address your concerns well?**
> >
> > Dear Reviewer ZXqE,
> >
> > Did we address your concerns well?
> >
> > - If yes, please would you consider raising your score appropriately?
> > - If not, please let us known which issues were not clarified sufficiently so that we have a chance to respond.
> >
> > Best regards,
> >
> > Authors

---

> > ### Author Response · Authors · 2021-08-19
> > **Reminder: Did we address your concerns well?**
> >
> > Dear Reviewer ZXqE,
> >
> > Did we address your concerns well?
> >
> > - If yes, please would you consider raising your score appropriately?
> > - If not, please let us known which issues were not clarified sufficiently so that we have a chance to respond.
> >
> > Best regards,
> >
> > Authors

---

### Official Review · Reviewer_xkpw · 2021-07-23

**Rating:** 6
**Confidence:** 4

**Summary:**

This paper proposed an error compensated based Katyusha method for finite-sum convex problems. The proposed method achieves the same asymptotic convergence rate as its full precision accelerated counterpart. Compared to another related work ADIANA, the proposed method can be applied to contractive compressor.

**Limitations And Societal Impact:**

Yes

**Main Review:**

The paper is clearly written and theoretically sound. However, I am not clear about the motivation of applying gradient compression to a finite-sum convex problems and what the real convex applications are that require gradient compression. Typically, the application requires gradient compression is the training of large deep neural networks, which is neither finite-sum nor convex. In addition, I have a few questions.

1. Theoretically, how do we select Q and Q_1 that minimizes the bound given a fixed communication budget?
2. What is the communication collective primitive used in the proposed algorithm? Since most of quantization and sparsification methods are incompatible with allreduce, it seems to me that it is all-to-all broadcast?
3. Since the proposed method obtains an accelerated rate that has better dependency on condition number, it is good to see the experimental results with varying condition numbers.
4. The proposed method introduces a number of hyper-parameters. How do we choose them in practice?

**Time Spent Reviewing:**

1

---

> ### Author Response · Authors · 2021-08-08
> **Issue 0: Gradient compression for convex problems?**
>
> > Issue 0:
>
> I am not clear about the motivation of applying gradient compression to a finite-sum convex problems and what the real convex applications are that require gradient compression. Typically, the application requires gradient compression is the training of large deep neural networks, which is neither finite-sum nor convex.
>
> > Reply to Issue 0:
>
> - Yes, typically, the best performing models in practice in many applications are nonconvex, as is the case with DNNs. However, while many companies may not like to admit that publicly as this does not create the right kind of PR these days, convex models such as logistic regression, for example, are used a lot, and one could argue that still even more so than nonconvex models. We have experience with working with companies such as Amazon, Google, Facebook and Apple - and in many applications, simple and light models such as logistic regression are used and effective. For example, Apple uses this (or at least used this about a year ago when we last checked) in their federated learning system that learns iPhone settings, which is a massive classification task. So, convex models are still relevant in practice as well. These models are large enough for communication compression to be very useful.
> - Many models have loss functions that are convex in the neighborhood of the (local or global) optimum. Understanding local convergence regime is also very important, and hence convexity assumptions can be also seen from this perspective: as an attempt to study local properties of nonconvex models.
> - Disregarding applications, we argue that the study of convex and strongly convex regime is absolutely essential from a scientific perspective. Indeed, if as a community we want to make progress on certain problems, we need to first understand them deeply in simpler scenarios. This is what we do in our paper. Error compensation and acceleration/momentum are widely used in isolation, both in the convex and nonconvex regimes. The question of whether these two techniques can be combined for a provably stronger performance than what can be offered by either one in isolation is thus very important. But it is a hard question, with no answer provided in the seven years from the time when Seide et al (2014) initially proposed error compensation. If there are some fundamental limits that do not allow such a combination, we should want to know. Lacking *any* results in this area, it is of course the right scientific approach to first study the problem in a simpler regime. This is standard practice, this is how science works. For example, a scientist may want to study these two tools for quadratic problems, or strongly convex problems, or for some other smaller class of problems first. Surely, if we can't solve the problem in a simpler setting, what chances do we have to solve it in a more complicated setting?
>
> Due to these reasons, we believe the criticism raised here is not valid.

---

> ### Author Response · Authors · 2021-08-08
> **Issue 1: Choosing $Q$ and $Q_1$**
>
> > Issue 1:
>
> Theoretically, how do we select $Q$ and $Q_1$ that minimizes the bound given a fixed communication budget?
>
> > Reply to Issue 1:
>
> - This question is hard in general. The reason for this is that there are infinitely many compressors that satisfy the properties these two compressors have in our theory (contractive (i.e., satisfying Assumption 3.1), and optionally satisfying Assumption 2.1), and our theory works for *all* such compressors. Moreover, these compressor classes are not yet fully understood, and new concrete compressors belonging to these families are still being discovered and proposed in papers. For many compressors we do not have explicit formulas for the compression ration (only bounds), and for some we do not have tight formulas for $\delta$ (only bounds). Optimizing over general classes of compressors which are not yet fully understood is not possible.
> - What can be done, however, is to pick a parametric subclass of compressors where the compression ratio offered by each compressor in the subclass is well understood, and optimize over the parameter. For example, we could pick the subclass of Top-$K$ compressors with varying $K$, and then optimize theoretical communication complexity over $K$. In this particular case we know what the compression ratio is, and we know $\delta$ as well. We shall include a simple example of this type in the paper; we agree this will be useful to the reader. Computations of this type are sometimes (but rarely) performed in papers. An example of such a computation, but for a different method (distributed SGD with unbiased bidirectional communication compression in the nonconvex regime; without any error compensation and without any acceleration), is given in Table 1 of the paper "Natural Compression for Distributed Deep Learning", 2019. We can perform something similar and add this to the paper.

---

> ### Author Response · Authors · 2021-08-08
> **Issue 2: Question about communication primitive used**
>
> > Issue 2:
>
> What is the communication collective primitive used in the proposed algorithm? Since most of quantization and sparsification methods are incompatible with allreduce, it seems to me that it is all-to-all broadcast?
>
> > Reply to Issue 2:
>
> - First of all, please note that our *algorithm* abstracts from what communication primitive is used in an actual implementation, which will be system-specific. In other words, the algorithm we analyze is independent of such considerations. Our method can in principle be applied to (cross-silo) federated learning where such primitives may not even be implemented, to commodity clusters, and even to supercomputers. All these compute systems differ in what communication primitives they support and how they are implemented. We do not build a system in this paper - our paper is of an algorithmic and theoretic nature - and hence we do consider these system specific issues.
> - Once our method is to be implemented on a particular compute system, one should of course take the constraints and capabilities of the system into consideration when choosing the various parameters our method supports (e.g., compressors). Indeed, when all-reduce is to be used, one needs to work with compressors that work well with this primitive. When the computers can communicate between each other directly (if each pair is connected via a direct link), then we do not need a centralized server, and all communication can be done in an all-to-all fashion, and any compressor can in principle be used. Again, we abstract from such considerations in this work. These are important considerations, but beyond the scope of our work, which simply has a different focus.

---

> ### Author Response · Authors · 2021-08-08
> **Issue 3: Experiments with varying condition numbers**
>
> > Issue 3:
>
> Since the proposed method obtains an accelerated rate that has better dependency on condition number, it is good to see the experimental results with varying condition numbers.
>
> > Reply to Issue 3:
>
> We *did* test the algorithms across a number of condition numbers. Perhaps you missed these plots? For example, see Figure 15 and Figure 16 in the Appendix (the default value of $\lambda$ is 1e-3).

---

> ### Author Response · Authors · 2021-08-08
> **Issue 4: How to choose the hyper-parameters?**
>
> > Issue 4:
>
> The proposed method introduces a number of hyper-parameters. How do we choose them in practice?
>
> > Reply to Issue 4:
>
> - As stated in Theorem 3.8, we can set all parameters properly (i.e., in a theory-supported way) if we know the Lipschitz constants, strong convexity parameter $\mu$, and the contraction factors for the compressors used.
> - Generally, we can estimate $\mu$ (from below) if an explicit strongly convex regularizer is used. Otherwise, this is not easy to estimate -and this is a well known problem in the literature of accelerate first order methods. Often, restarting techniques help to deal with this issue.
> - A lower bound for the contraction parameter $\delta$ can often be obtained easily, and for most compressors used in practice, these values are known. We have given some examples in the paper.
> - Lipschitz constants can often be estimated in the standard way as this is done in all other first-order methods. This is not a problem specific to our work, and hence we do not feel the need to elaborate on this.
> - One can search for the optimal stepsize as we do in our experiments. Noticing that the performance of w6a (a5a) is similar to that of w8a (a9a), this indicates that we can do the optimal stepsize search in a small sample of the data set if it is very large in practice.
>
> In general, since our method is supposed by strong theory which says how the various parameters should be chosen, we are in a far better shape when deciding on the hyper parameters than we would have been if our method was just a heuristic, even if no closed formulas for these parameters exist. Theory narrows down  the search space dramatically, and one can always do some fine-tuning for better performance.

---

> ### Author Response · Authors · 2021-08-08
> **All issues handled**
>
> Dear reviewer,
>
> Thanks for the positive evaluation of our paper. We have responded to all four of your concerns. Here is a quick summary:
>
> - Issue 0: We hope we explained that this is not an issue.
> - Issue 1: In general, what you ask is not possible, but in a more narrow sense, this can be done, and will do it.
> - Issue 2: We believe this question is unrelated to the contributions of our paper, as we explain.
> - Issue 3: Not an issue since we actually have such experiments in our paper; seems you missed them.
> - Issue 4: We explained how the hyper-parameters can be chosen. We have theoretical expressions for all, and many can often be easily computed or estimated. Of course, sometimes there are difficulties.
>
> If you agree that we managed to address all issues, please consider raising your mark. If you believe this is not the case, please let us know so that we have a chance to respond.
>
> Thank you!
>
> Authors

---

> > ### Author Response · Authors · 2021-08-15
> > **How did we do in our rebuttal?**
> >
> > Dear Reviewer xkpw,
> >
> > Did we address all 5 concerns satisfactorily?
> >
> > - If yes, please would you consider raising your score appropriately?
> > - If not, please let us known which issues were not clarified sufficiently so that we have a chance to respond.
> >
> > Best regards,
> >
> > Authors

---

> > ### Author Response · Authors · 2021-08-19
> > **Reminder: How did we do in our rebuttal?**
> >
> > Dear Reviewer xkpw,
> >
> > Did we address all 5 concerns satisfactorily?
> >
> > - If yes, please would you consider raising your score appropriately?
> > - If not, please let us known which issues were not clarified sufficiently so that we have a chance to respond.
> >
> > Best regards,
> >
> > Authors

---

### Official Review · Reviewer_5Tji · 2021-07-24

**Rating:** 7
**Confidence:** 4

**Summary:**

The authors consider accelerated training algorithms using gradient compression with error compensation. For this purpose, they incorporated gradient compression and error compensation in the Katyusha algorithm and developed ECKL. The algorithm quantizes both the SG and VR term and keeps track of the quantization error for error compensation in the next iteration.

The paper analyzed the convergence rate and iteration complexity of ECKL for the smooth convex functions and compared it with variants of DIANA both theoretically and via some simple simulations on logistic regression.

**Main Review:**

The idea and algorithm developed in the paper is interesting and can be potentially useful in distributed DNN training. However, one shortcoming of the paper is its analysis.The paper only considers the class of smooth convex functions. Although for theoretical analysis, the assumptions in the paper are common and ok, but for the simulations, it would be more interesting to see how the algorithm performs for more complex cost functions, even training a simple DNN on CIFAR10 could have demonstrated the performance of ECKL in more general setting.

Minor:
Finally, the writing and notations of the paper can be improved. Some notations are used without proper definition, and some parts are missing, e.g.,
* what is parameter $\mathcal{L}_1$?
* Algorithm 1, how $h_\tau^0$ is initialized?
* Alg. 1: $u_\tau^{k+1}$ is always set to 0 for $\tau=2,\ldots,n$, and only $u_1^k$ is changing and used. It seems that worker 1 acts as the controller to decide on updating $\omega^k$. In that sense, defining $u_\tau^k$ looks redundant and a single shared random variable $u^k$ would be enough.

____________________________
UPDATE:
Thanks for the answers and clarifications. I updated the my recommendation accordingly. Although not necessary and having simulations on convex problems suffices for the paper, I believe that showing the performance on some non-convex problems can really add to the value of the paper.

**Time Spent Reviewing:**

4

---

> ### Author Response · Authors · 2021-08-08
> **Issue 1: Experiments with DNNs**
>
> > Issue 1:
>
> The idea and algorithm developed in the paper is interesting and can be potentially useful in distributed DNN training. However, one major shortcoming of the paper is its analysis.The paper only considers the class of smooth convex functions. Although for theoretical analysis, the assumptions in the paper are common and ok, but for the simulations, it would be more interesting to see how the algorithm performs for more complex cost functions, even training a simple DNN on CIFAR10 could have demonstrated the performance of ECKL in more general setting.
>
> > Reply to Issue 1:
>
> Please note that the nonconvex regime is simply out of the scope of this paper, both theoretically, and computationally. Naturally, each algorithm has its space of applicability, and it does not make sense to apply it outside of this realm. A sorting algorithm is not designed to perform facility location, and clustering method is not designed to perform binary classification, and Nesterov's accelerated gradient method is not designed to find stationary points or global minima of nonconvex functions. This is natural.
>
> In analogy to this, our method ECLK is specifically designed to solve distributed optimization problems in the strongly convex regime (in a communication efficient manner, using a combination of acceleration and error compensation, which was never achieved before). ECLK relies on strong convexity, and may not work if strong convexity is not present. This is not a deficiency of our method. It is simply an honest acknowledgement that each method has its natural limitations. It is well known that there is a very sharp divide between acceleration in the strongly convex and nonconvex regimes, even in non-distributed settings without any gradient compression. Algorithmic approaches to these two problems are dramatically different, and he type and amount of acceleration one can achieve is also different. For example, finite sum problems with subsampling, the SARAH/SPIDER/PAGE methods are optimal in the smooth nonconvex regime. The acceleration mechanism in these methods does not resemble Nesterov/Katyusha momentum - and instead is based on carefully designed sequence of biased stochastic gradient estimates. The situation is very different in the strongly convex regime.
>
> Since our method was specifically designed for strongly convex problems, and since it relies on parameters characterizing the degree of strong convexity, all we can do if we are to try to train DNNs with it is to run the method heuristically, "hoping" for the best. But there is no reason to expect it to do well, and hence there is not much such an experiment will add to our paper.
>
> We believe that since we solved a major open problem in the literature on error compensation, and performed numerical experiments on problems that the method was designed to solve with promising results, we have done good justice to the topic we are studying.
>
> ---
>
> *Comment:* In general there is an expectation among some reviewers to test every new ML method on DNNs. This is not a good trend. Not all methods are designed to train DNNs, and not all research is related to DNNs. As a community, we need to understand that the world of research goes far beyond the current hot topics.

---

> ### Author Response · Authors · 2021-08-08
> **Issue 2: Minor Comments**
>
> > Issue 2:
>
> Minor: Finally, the writing and notations of the paper can be improved. Some notations are used without proper definition, and some parts are missing, e.g.,
> - what is parameter ${\cal L}_1$?
> - Algorithm 1, how $h_\tau^0$ is initialized?
> - Alg. 1: $u_\tau^{k+1}$ is always set to 0 for $\tau=2, \dots, n$, and only $u_1^k$ is changing and used.  It seems that worker 1 acts as the controller to decide on updating $\omega^k$. In that sense, defining $u_\tau^k$ looks redundant and a single shared random variable $u^k$ would be enough.
>
> > Reply to Issue 2:
>
> We agree these are minor issues:
> - ${\cal L}_1$ is a positive constant used as a parameter in Algorithm 1. We set the value for ${\cal L}_1$ in Theorem 3.8. We will mention this again when describing the method so that a reader is less likely to miss this.
> - $h_\tau^0$ can be any vector in $\mathbb{R}^d$; our theory works for all initializations. We will clarify this in the paper.
> - Yes, we can also use a single shared random variable $u^k$ instead. Again, we will add a sentence clarifying this.

---

> ### Author Response · Authors · 2021-08-08
> **All issues handled**
>
> Dear reviewer,
>
> Thanks for the positive evaluation of our paper. We have responded to all four of your concerns. Here is a quick summary:
>
> - Issue 1, your main concern, is not an issue. In fact, we believe reviewers should not be asking for methods designed to solve problems from category A (strongly convex problems) to solve problems in category B (nonconvex problems), and view the omission of such experiments as a major issue. We would be very happy if other reviewers and the AC could voice their view on this as well. We could add such experiments if you insist, but in principle this really should not be necessary nor desired, as this does not add scientific value to our paper.
> - Issue 2 is a collection of very minor issues each of which can be handled by adding 1-2 sentences in appropriate places in the text.
>
> If you agree that we managed to address all issues, please consider raising your mark. If you believe this is not the case, please let us know so that we have a chance to respond.
>
> Thank you!
>
> Authors

---

> > ### Author Response · Authors · 2021-08-15
> > **Did we answer all your questions to your satisfaction?**
> >
> > Dear Reviewer 5Tji,
> >
> > Did we address both of your concerns satisfactorily?
> >
> > - If yes, please would you consider raising your score appropriately?
> > - If not, please let us known which issues were not clarified sufficiently so that we have a chance to respond.
> >
> > Best regards,
> >
> > Authors

---

> > ### Author Response · Authors · 2021-08-19
> > **Reminder: Did we answer all your questions to your satisfaction?**
> >
> > Dear Reviewer 5Tji,
> >
> > Did we address both of your concerns satisfactorily?
> >
> > - If yes, please would you consider raising your score appropriately?
> > - If not, please let us known which issues were not clarified sufficiently so that we have a chance to respond.
> >
> > Best regards,
> >
> > Authors

---

### Official Review · Reviewer_XZRR · 2021-07-28

**Rating:** 7
**Confidence:** 4

**Summary:**

This paper presents accelerated methods for communication-efficient learning applications. In particular, a loopless Katyusha strategy is introduced to attain the O(\sqrt{L/\mu}) rate for error-compensated methods using stochastic gradient information. These methods are called ECLK. In terms of theoretical iteration complexity and empirical experiments, ECLK is explicitly shown to be more communication efficient and to have a better rate than existing error compensation methods.

**Ethical Concerns:**

Ethical concerns in terms of methodology are addressed by the authors. For instance, python codes are provided for the reviewers to re-run the experiments on the benchmark data where its source was available and properly cited.

**Ethics Review Area:**

["Discrimination / Bias / Fairness Concerns", "Inadequate Data and Algorithm Evaluation"]

**Limitations And Societal Impact:**

The contribution of this paper is primarily theoretical. Therefore, limitations and societal impacts are not applicable.

**Main Review:**

This paper is clearly written. The simulation results highlight the impact of algorithmic tuning parameters, and the superior performance of ECLK over non-accelerated error-compensated algorithms. However, there are some concerns presented below:

* The authors claim that ECLK performs better than EC-LSVRG-DIAN but worse than ADIANA, in theoretical iteration complexity. However, to compare these complexities, Lyapunov functions associated with these methods are totally different. The Lyapunov function for ECLK is the summation of the Euclidean distance and the function difference, while the Lyapunov function for EC-LSVRG-DIANA and ADIANA is only the function difference. Therefore, comparing the complexities with different Lyapunov functions seems to weaken the authors' claim.
* Literature review related to this work is in Section F in the supplementary materials. This approach leads to weak discussions of error compensation in the introduction. This literature review in Section F should be included rather in the introduction.
* This paper lacks the motivation on why Katyusha is used as the acceleration technique, instead of Nesterov's acceleration. It will be better if the authors also include literature review about this technique.
* One section in numerical experiments highlights that ECLK is comparable to ADIANA that uses non-compensated information for most cases in deterministic settings. This raises the question why the authors instead consider the extension of ADIANA for error-compensated methods. In addition, what happens in stochastic settings? Since Katyusha, unlike Nesterov's acceleration, guarantees the fast rate for the methods to solve stochastic optimization, ECLK may be expected to outperform ADIANA and error-compensated algorithms with Nesterov's acceleration. This will strengthen the authors' contributions.

**Time Spent Reviewing:**

4

---

> ### Author Response · Authors · 2021-08-08
> **Issue 1: Comparison to EC-LSVRG-DIANA and ADIANA**
>
> >Issue 1:
>
> The authors claim that ECLK performs better than EC-LSVRG-DIANA but worse than ADIANA, in theoretical iteration complexity. However, to compare these complexities, Lyapunov functions associated with these methods are totally different. The Lyapunov function for ECLK is the summation of the Euclidean distance and the function difference, while the Lyapunov function for EC-LSVRG-DIANA and ADIANA is only the function difference. Therefore, comparing the complexities with different Lyapunov functions seems to weaken the authors' claim.
>
> >Reply to Issue 1:
>
> As we explain below, the above is not an issue. Please let us know if this explanation settles your concern!
>
> A) While the ADIANA complexity result in their main theorem (Theorem 4) is stated in terms of distance to optimal solution only,
>
> $$|| z^k - x^* ||^2,$$
>
> the proof behind their result depends on a rather complicated Lyapunov function that is not entirely different from ours. Indeed, Li et al in their ADIANA work show (see their equation (14)) that
>
> $$
> \mathbb{E}\left[c_{1} \mathcal{Y}^{k+1}+c_{2} \mathcal{Z}^{k+1}+c_{3} \mathcal{W}^{k+1}+c_{4} \mathcal{H}^{k+1}\right] \leq\left(1-c_{5}\right)\left(c_{1} \mathcal{Y}^{k}+c_{2} \mathcal{Z}^{k}+c_{3} \mathcal{W}^{k}+c_{4} \mathcal{H}^{k}\right),
>  $$
>
> for some positive constants $\{c_i\}$, where
>
> $$
> \mathcal{Z}^{k} = ||z^{k}-x^* ||^{2}, \quad {\cal Y}^{k} =P (y^{k})-P(x^*), \quad {\cal W}^{k} =P(w^{k})-P(x^*), \quad \mathcal{H}^{k} =\frac{1}{n} \sum_{i=1}^{n} || h_{i}^{k} - \nabla f_{i} (w^{k}) ||^{2},
> $$
> and $\{z^k\}, \{y^k\}, \{w^k\}, h_i^k$ are  iterates produced by ADIANA. You will note that our Lyapunov function contains very similar terms, but also additional terms such as
>
> $$ \frac{1}{n} \sum_{\tau=1}^n || e_{\tau}^k ||^2$$
>
> that is needed because of error compensation. So, it is *not true* that the Lyapunov functions for ECLK and ADIANA are entirely different.
>
> B) Moreover, note that one can lower bound both of these Lyapunov functions by any of the terms comprising them as they are all nonnegative. So, a rate that applies to the Lyapunov functions all applies to all individual terms! In particular, it applies to function suboptimality for both ECLK and ADIANA. This is also the expression that is bounded in the convergence theorem for EC-LSVRG-DIANA. So, in contrast to what the reviewer says, the rates in fact *are comparable*.
>
> C) One of the advantages of a Lyapunov style analysis is that one obtains convergence of *all* terms constituting the Lyapunov function, not merely of standard objects, such as iterates or function suboptimality. In the case of ECLK, we also obtain the convergence of the compression error $e_\tau^k$ and of the learning vector $h_\tau^k$, which shows that our method indeed provably performs error compensation, and how. In this sense, our results are slightly stronger than what they would have been if we merely showed convergence for function suboptimality, for example.
>
> We will clarify these points in the paper - this can be done by adding a few sentences.

---

> ### Author Response · Authors · 2021-08-08
> **Issue 2: "Literature review" should be moved from supplementary material to the introduction**
>
> >Issue 2:
>
> Literature review related to this work is in Section F in the supplementary materials. This approach leads to weak discussions of error compensation in the introduction. This literature review in Section F should be included rather in the introduction.
>
> >Reply to Issue 2:
>
> As we argue below, this is a very minor issue if this can be seen as an issue at all.
>
> Please note that Section F is *not* the "Literature Review" part of our paper, as the reviewer is trying to suggest. We call section "Related Work: Extra Comments". This section contains some extra comments that *complement* the literature review we already have in the introduction and contribution sections. Please see that indeed, we perform literature review of the most closely related pieces of work in the main paper.
>
> Moreover, we believe it does not really matter much where the text which we decided to include in Section F is present - and hence this is a very minor comment that should not in any affect the score of our paper. We moved this somewhat less relevant part of literature review to the appendix as this allowed us to get extra space for an outline of the proof of our main result via the statement of the lemmas that we needed to prove to obtain it. We believe this was a good trade as it is more important to describe actual contributions in a page-limited paper than to use excessive space to perform a a more thorough literature review. 9 pages of space provide for a zero-sum game for content...
>
> Should our paper be accepted, and we hope it will, we will get an extra page in the camera ready version of the paper, and we will be most happy to move this part to the introduction.

---

> ### Author Response · Authors · 2021-08-08
> **Issue 3: Why Katyusha acceleration rather than Nesterov acceleration?**
>
> > Issue 3:
>
> This paper lacks the motivation on why Katyusha is used as the acceleration technique, instead of Nesterov's acceleration. It will be better if the authors also include literature review about this technique.
>
> >Reply to Issue 3:
>
> This is a minor point that can be handled by adding a couple clarifying sentences to the paper.
>
> The acceleration in Katyusha or loopless Katyusha is an extension of Nesterov's acceleration to make it work in the stochastic regime. In contrast to Nesterov acceleration, which is designed to work for full-batch gradient methods, Katyusha uses a negative momentum technique. Since our method performs (possibly randomized) compression, and also uses subsampling on each node, we need to rely on this type of momentum. We rely on the loopless version of Katyusha as that is easier to analyze, and this bit of extra simplicity is needed for us to deal with the complications brought in by biased gradient estimators and error compensation.
>
> We will clarify this in the camera ready version of the paper.
>
> Thanks for this suggestion!

---

> ### Author Response · Authors · 2021-08-08
> **Issue 4: ADIANA for error-compensation?**
>
> > Issue 4:
>
> One section in numerical experiments highlights that ECLK is comparable to ADIANA that uses non-compensated information for most cases in deterministic settings. This raises the question why the authors instead consider the extension of ADIANA for error-compensated methods. In addition, what happens in stochastic settings? Since Katyusha, unlike Nesterov's acceleration, guarantees the fast rate for the methods to solve stochastic optimization, ECLK may be expected to outperform ADIANA and error-compensated algorithms with Nesterov's acceleration. This will strengthen the authors' contributions.
>
> > Reply to Issue 4:
>
> We believe the suggestion made here by the reviewer does not make sense, and comes from a misunderstanding by the reviewer about what ADIANA method is and does.
>
> Let us explain.
>
> ADIANA is an accelerated version of the DIANA algorithm proposed by Mishchenko et al (2019). DIANA can be seen as an "error-compensation-type" technique specifically designed for unbiased compressors. The DIANA technique only works for unbiased compressors and does not extend to general contractive compressors. It is very efficient as it can take advantage of the unbiasedness and independence of the compressed gradient vector across the worker nodes, which reduces variance.
>
> The class of contractive compressors we consider in our paper can be seen as a superset of the above mentioned class of unbiased compressors. Indeed, as we explain in the paper, any unbiased compressor can be turned into a contractive compressor via appropriate scaling. However, not all contractive compressor arise in that way, and example being the Top-$K$ compressor. Because of this, it is naturally harder to fix/compensate the error caused by general contractive compressor (unless, of course, we know they arise as scaled version of unbiased compressors, in which case we can handle them via the DIANA mechanism).
>
> A mechanism that tries to achieve for the general class of contractive compressor what DIANA achieves for the smaller class of unbiased compressors is called Error Compensation. However, this mechanism is much less understood. This is mainly because in the analysis we cannot rely on unbiasedness and independence of the compressors across the devices, and the lack of these properties leads to severe complications.
>
> - Let us illustrate this on one example: While DIANA achieves *linear rate* for strongly convex problems with standard assumptions, error compensation does not. Existing error compensation techniques achieve a linear rate for over-parameterized models (characterized by  $\nabla f_{\tau}(x^*) = 0$ for all devices $\tau$), but not in general. [This was recently fixed by the EC-LSVRG-DIANA method with the help of additional unbiased compressors.]
>
> - Another key difference, and this is key to our paper, is the ability of these two techniques to work in tandem with *acceleration*. While DIANA can be provably accelerated, and this was achieved by the ADIANA method, there is no accelerated version of error compensation. This is precisely what we address in our paper, and thereby resolve an important open problem in the literature on error compensation.
>
> We hope this sheds light on the "philosophical" connection between ECLK and ADIANA. Let us recap: In the realm of non-accelerated methods, we have two techniques for handling the error/variance induced by compression:
> - DIANA for unbiased compressors
> - Error Compensation for contractive compressors
>
> *ECLK accelerates Error Compensation in an analogous way in which ADIANA accelerates DIANA.*
>
> In the light of this, please note that it is not correct to say, as the reviewer does, that "ADIANA uses non-compensated information". ADIANA is an accelerated version of DIANA, which successfully compensates for the errors caused by the independent unbiased compressors it is based on. And ADIANA inherits this property.
>
> It should be apparent from the above that to us it does not make sense to "consider the extension of ADIANA for error-compensated methods", as the reviewer suggests. We do not see how this idea makes sense from our point of view as experts in these types of methods. If the reviewer still believe this is a good idea, then please can you elaborate on what do you mean *exactly*, in more detail? Of course, we will be happy to consider any good suggestions! In case we misunderstood what the reviewer wanted to say, please can you elaborate more so that your message/suggestion gets across to us?
>
> More comments:
> - Since ADIANA relies on the assumption that full/exact gradient is computed on all nodes, to be fair, we naturally compare ADIANA with ECLK-F ( = ECLK which also uses full gradients); and we do so in Figure 4.
> - However, we also compare the stochastic variant of ECLK (i.e., using stochastic gradient for ECLK) with ADIANA in Figure 11 and Figure 15 in the appendix.

---

> ### Author Response · Authors · 2021-08-08
> **All issues handled**
>
> Dear reviewer,
>
> Thanks for the positive evaluation of our paper. We have responded to all four of your concerns. Here is a quick summary:
> - Issue 1 is not an issue, as we explain.
> - Issue 2 is a minor recommendation at best, and can be easily handled.
> - Issue 3 is minor as it can be handled by a few sentences of extra explanation.
> - Issue 4 is not an issue and it seems to be based on a misunderstanding by the reviewer on the relationship between DIANA, ADIANA, Error compensation and ECLK.
>
> If you agree that we managed to address all issues, please consider raising your mark. If you believe this is not the case, please let us know so that we have a chance to respond.
>
> Thank you!
>
> Authors

---

> > ### Author Response · Authors · 2021-08-15
> > **Did we address all your concerns satisfactorily?**
> >
> > Dear Reviewer XZRR,
> >
> > Did we address all your concerns satisfactorily?
> >
> > - If yes, please would you consider raising your score appropriately?
> > - If not, please let us known which issues were not clarified sufficiently so that we have a chance to respond.
> >
> > Best regards,
> >
> > Authors

---

> > ### Author Response · Authors · 2021-08-19
> > **Reminder: Did we address your concerns satisfactorily?**
> >
> > Dear Reviewer XZRR,
> >
> > Did we address all your concerns satisfactorily?
> >
> > - If yes, please would you consider raising your score appropriately?
> > - If not, please let us known which issues were not clarified sufficiently so that we have a chance to respond.
> >
> > Best regards,
> >
> > Authors

---

### Author Response · Authors · 2021-08-08
**Thanks to all reviewers**

We wish to thank the reviewers for examining our manuscript, and investing time in the review process.

We thank them all for appreciating our work in this way:

- "This paper is clearly written." (Reviewer XZRR)
- "In terms of theoretical iteration complexity and empirical experiments, ECLK is explicitly shown to be more communication efficient and to have a better rate than existing error compensation methods." (Reviewer XZRR)
- "The simulation results highlight the impact of algorithmic tuning parameters, and the superior performance of ECLK over non-accelerated error-compensated algorithms." (Reviewer XZRR)
 - "The idea and algorithm developed in the paper is interesting and can be potentially useful in distributed DNN training." (Reviewer 5Tji)
- "The paper is clearly written and theoretically sound." (Reviewer xkpw)
- "The paper is clearly written and well organized." (Reviewer ZXqE)
- "...the theoretical contributions of the paper seem very relevant to the machine learning and stochastic optimization communities" (Reviewer iBtS)

While some issues were raised, we believe that no *major* criticism was raised. As you will see in our response, we are able to handle all issues. Some are based on a simple misunderstanding, and others are minor, and can hence be easily addressed. No issues related to the validity of the theory, nor superiority of the rates were raised.

We wish to reiterate here a paragraph from our paper, page 2:

>> One of the key open problems in this area—the problem we address in this paper in the affirmative— is whether it is possible to design provably accelerated gradient- type methods, in the sense of Nesterov [Nesterov, 1983, 2004], that work with contractive compressors.

Our work is the first that solves this open problem, and hence we believe our work is a breakthrough in the area of error-compensated distributed methods, which were first proposed in 2014 by Seide et al.

We hope that you will engage with us in a back-and-forth discussion. We will be most happy to answer any remaining questions!

Best regards,

Authors

---

### Decision · Program_Chairs · 2021-09-27

**Decision:**

Accept (Poster)

**Comment:**

Overall, this paper addresses an important open problem. It is well-written, and solid theoretical results are complemented by distinct advantages in simulation experiments.

Based on my own reading, and the comments of the reviewers, I recommend acceptance.

There are some weak points that should be improved in the camera ready version.

The abstract and part of the introduction are a little bit misleading, referring to Nesterov’s acceleration (easily mistaken for Nesterov’s accelerated gradient method), while the algorithm later uses Katyusha (another acceleration technique).

The writing is a bit too dense, with little exposition on how the algorithm works and how it is related to previous work. In the response to the reviewers, you claim that this is not an issue and it does not matter if you put material in the main text or the appendix. I disagree. The main text should be complete and accessible to a broader audience than researchers in error-compensated training algorithms. I believe that parts of the related work from the appendix need to be moved into the main text and that you need to expand on how the algorithm works, and how its parameters are selected. In the response to the reviewers, you share such intuition and describe parameter tuning techniques. This knowledge should also make it into the main document.

Finally, I also believe that you can clarify in what environment you perform your experiments. I see the main contribution of this paper as mainly theoretical, but you do talk about “nodes” in your experiments, and “send” and “receive” in your algorithm. To me, it is perfectly fine if you have made the experiments in a Python/Matlab/C++… simulation where send and receive operations are instantaneous and error-free. But you should be clear with this in the paper.